# Label-free mass and size characterization of few-kDa biomolecules by hierarchical vision transformer augmented nanofluidic scattering microscopy

Henrik K. Moberg[1], Bohdan Yeroshenko[1], Joachim Fritzsche [1],
David Albinsson [2], Barbora Spackova[3], Daniel Midtvedt [4],
Giovanni Volpe [4] & Christoph Langhammer [1] ✉

Nanofluidic scattering microscopy characterizes single molecules in sub-wavelength nanofluidic channels label-free, using the interference of visible light scattered by the molecule and nanochannel. It determines a molecule's hydrodynamic radius by tracking its diffusion trajectory and its molecular weight by analyzing its scattering intensity along that trajectory. However, using standard analysis algorithms, it is limited to characterization of proteins larger than $\approx 60$ kDa. Here, we push this limit by one order of magnitude to below $\approx 6$ kDa molecular weight and $\approx 1.5$ nm hydrodynamic radius − as we exemplify on the peptide hormone insulin − by using ultrasmall nanofluidic channels and by analyzing the data with a hierarchical vision transformer. When we benchmark this approach against the theoretical limit set by the Cramér−Rao Lower Bound, we find that it can be approached with sufficiently long molecular trajectories. This enables quantitative label-free single-molecule microscopy for biologically relevant families of sub-10-kDa molecules, such as cytokines, chemokines and peptide hormones.

Label-free single-molecule microscopy is a growing field that has advanced rapidly with increasingly sophisticated methodologies, such as interferometric scattering microscopy (iSCAT)[1–6], plasmonics[7–12], plasmonic scattering microscopy[13], evanescent scattering microscopy[14], single-protein oscillators[15], dielectric resonators[16–18], Fabry-Pérot microcavities[19], and our own recently introduced nano-fluidic scattering microscopy (NSM)[20]. This development stems from the limits of fluorescence, as labels can alter function and require prior knowledge of targets[21,22]. Within this landscape, NSM is positioned among label-free single-molecule approaches that do not require surface immobilization during measurement, enabling the character-ization of freely diffusing analytes in solution while preserving native behavior.

Simultaneously, in all existing label-free single-molecule microscopy methods, the signal-to-noise ratio decreases with the size or mass of the analyte to be studied. Consequently, these methods either cannot access at all biologically relevant families of molecules in the sub-10 kDa size regime[20,23–27], or they can detect but not accurately characterize them quantitatively in terms of molecular weight (MW) and/or hydrodynamic radius ($R_s$)[19,28].

In the last decade, in parallel to label-free single-molecule microscopy, deep-learning-based computer vision algorithms have also progressed quickly. Alongside their widespread application in autonomous vehicles, image-based medical diagnostics, and computer vision algorithms have also gained traction in optical microscopy, e.g.,

[1]Department of Physics, Chalmers University of Technology, Gothenburg, Sweden. [2]Envue Technologies, Gothenburg, Sweden. [3]Institute of Physics of the Czech Academy of Sciences, Prague, Czechia. [4]Department of Physics, Gothenburg University, Gothenburg, Sweden. ✉e-mail: clangham@chalmers.se

in virtual super-resolution[29], chemical staining[30], and analysis of biological samples[31–33].

To leverage this development, we introduce a hierarchical vision transformer (h-ViT) algorithm that we have specifically developed for NSM that enables real-time label-free detection and MW and $R_s$ characterization of single molecules diffusing inside a nanochannel[20]. We demonstrate the capabilities of h-ViT in NSM data analysis by (i) studying a 200, 100, and 50 bp DNA ladder with nominal MW of 121/132, 61/66 and 30/33 kDa, depending on the calculation method[34,35], in a single nanochannel with cross-sectional area $A_I$ = 122 nm × 97 nm, and by (ii) determining the peptide hormone Insulin's MW to 5.95 ± 0.16 kDa and $R_s$ to 1.52 ± 0.13 nm inside a small nanochannel $A_{II}$ = 63 nm × 30 nm. These obtained limits of detection (LoDs) improve on previously reported limits of approximately 60 kDa MW and 3.9 nm $R_s$ for NSM[20], and also in general for label-free microscopy workflows that go beyond single molecule detection, i.e., enable the characterization of molecular properties like MW or $R_s$. We benchmark these results by comparing them to the theoretical limit set by the Cramér–Rao Lower Bound (CRLB) and find that h-ViT converges towards this theoretical limit for molecular trajectories >10,000 frames, which is achievable with NSM.

To briefly also put this work into an application perspective, we note that the MW determination of single molecules enabled by NSM[20] is equivalent to mass photometry in iSCAT[36] but differs in that the studied molecules diffuse freely in solution and are not potentially altered by a specific binding interaction with a surface, provided that also non-specific binding is avoided, e.g., by repulsive electrostatic interactions or appropriate antifouling coatings. As a second

difference, NSM also provides information about $R_s$. Consequently, the NSM method can be used in the determination of homogeneity, oligomeric state, and stoichiometry of biological samples, e.g., in drug development or genomic DNA analysis, or in studies of molecular interactions in solution to estimate binding affinities or aggregation and degradation. Furthermore, NSM can be applied to characterize biological nanoparticles, such as extracellular vesicles[20] or lipid nanoparticles and their molecular cargos, such as mRNA, and thus be used in similar contexts as dynamic light scattering (DLS) or nanoparticle tracking analysis (NTA) with the ability to resolve smaller nanoparticles. When it comes to the specific application of NSM on single molecules in the sub-10 kDa regime addressed in the present work, it enables the study of molecular systems such as cytokines, chemokines, or peptide hormones that are biologically highly relevant but have remained elusive for label-free microscopy techniques due to their small sizes.

## Results

### Nanofluidic scattering microscopy

The core of NSM is a 1 × 1 cm² fluidic chip that features microfluidic in- and outlet systems connected to nanofluidic channels tailored for the study of specific biomolecules in the NSM experiment, and to reservoirs that interface a custom chip holder (Fig. 1A, B). The fluidic system is nanofabricated into the thermal oxide of a silicon wafer and sealed with a bonded glass lid. It can be pressurized by $N_2$ gas to enable convective analyte flow from a reservoir towards and through the nanochannels to prepare an experiment. During the NSM measurement itself, the flow is turned off to let molecules freely diffuse.

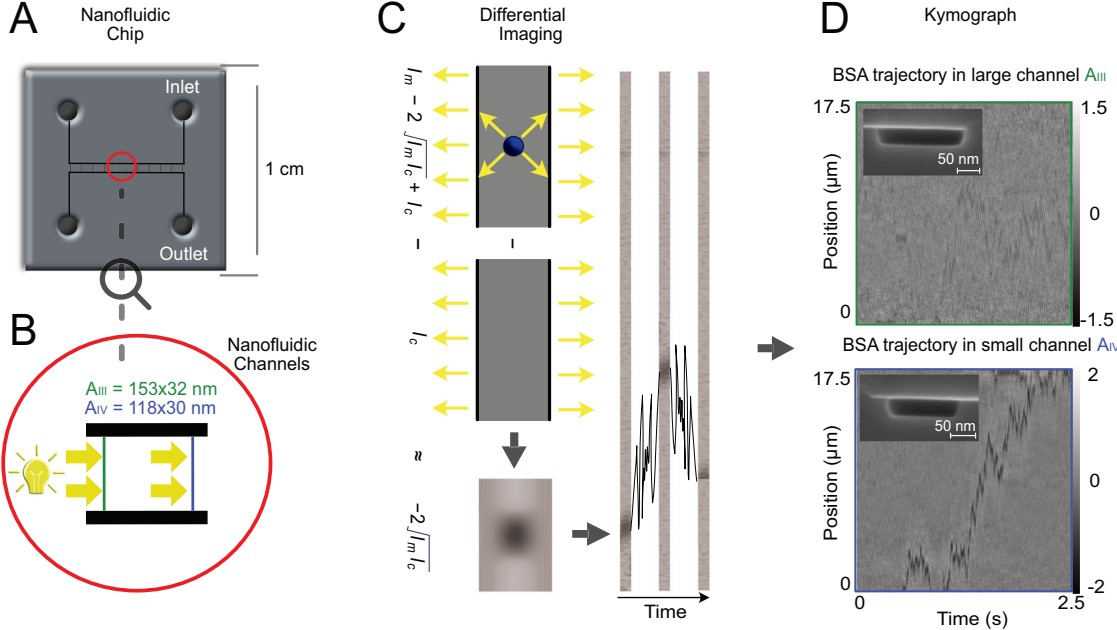

**Fig. 1 | The NSM principle and the impact of nanochannel cross section area.**
**A** Schematic of the nanofluidic chip used. **B** Zoom-in on the nanofluidic part of the chip depicting the microfluidic inlet and outlet channels that connect to liquid reservoirs used to interface the chip holder and to a set of parallel nanofluidic channels used for NSM experiments. The length of the nanochannels is chosen such that they fit the field of view of the microscope at the desired magnification, and the cross-section of the nanochannels is tailored to the size of the molecule to be analyzed, as discussed in the main text. The two indicated channel dimensions correspond to the two nanochannel types used in the experiment depicted in (**D**). **C** The principle of differential imaging in NSM, in which we subtract the light scattered (yellow arrows indicate the scattered-light direction) by an empty nanochannel from the light scattered by the same channel with a molecule inside. A sequence of differential images of a nanochannel containing a diffusing single molecule obtained in this way is combined into a kymograph in (**D**), which then

contains the full molecular trajectory. Here, this is exemplified for a Bovine Serum Albumin (BSA, MW = 66.8 kDa, $R_s$ = 3.5 nm) molecule differentially imaged in two nanochannels with different cross sections (defined by the widest and deepest points of the cross sections; see insets for cross-section scanning electron microscope images), i.e., and $A_{III}$ = 153 nm × 32 nm and $A_{IV}$ = 118 nm × 30 nm. While the trajectory of the BSA molecule is not resolved for the larger $A_{III}$ channel when only applying the standard data preprocessing steps outlined in the Methods section, it is clearly visible in the smaller channel $A_{IV}$. This showcases the potential of lowering the LoD of NSM by reducing the nanochannel cross section area $A$, since $A$ is inversely proportional to the LoD[20]. For a more detailed statistical analysis of BSA and its dimeric oligomers, we refer to our seminal NSM work[20]. Here, NSM denotes nanofluidic scattering microscopy, BSA bovine serum albumin, MW molecular weight, $R_s$ hydrodynamic radius, and LoD limit of detection.

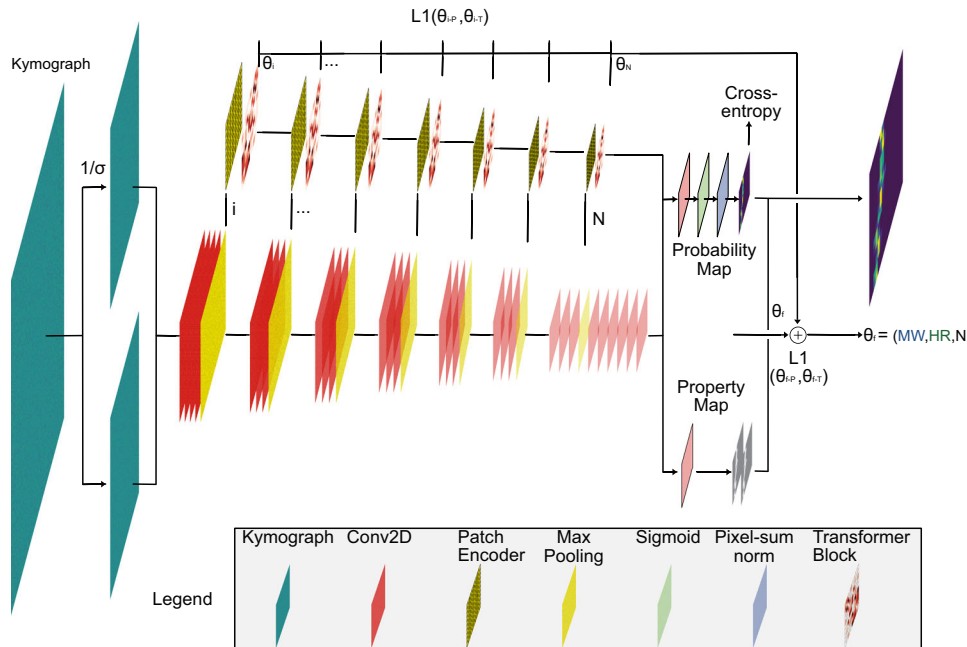

**Fig. 2 | Hierarchical vision transformer (h-ViT) for single biomolecule characterization by NSM.** The h-ViT model processes NSM kymographs to predict the MW and $R_s$ of single particles, such as biomolecules. Initially, Conv2D layers extract spatial features, followed by max pooling for downsampling. The Patch Encoder then encodes these feature maps into smaller patches, while retaining crucial spatial information. Each patch is passed through Transformer Blocks, where multi-head attention mechanisms selectively focus on different regions of the input, capturing long-range dependencies in the kymograph. The outputs of the Transformer Blocks are processed to generate two key outputs: a Probability Map (indicating the likelihood of particle trajectories at different locations) and a Property Map (predicting specific biomolecule properties, such as MW, $R_s$, and the number of trajectory points $N$). The model utilizes cross-entropy loss for classification tasks and L1 loss to minimize the error between predicted and true molecular properties. The estimation of each parameter ($\theta_i$) representing MW and $R_s$, and the number of trajectory points, $N$, is refined through iterative back-propagation. Finally, the model weighs the Property Map with the Probability Map to output a final estimation of the biomolecule properties, weighted by the probability of said biomolecule being present in any given region. We note that the coarse resolution along the nanochannel direction is the consequence of h-ViT looking at 7 scales of downsampling of kymographs that are 512 pixels long (i.e., $512/2^7 = 4$). Here, NSM denotes nanofluidic scattering microscopy, h-ViT hierarchical Vision Transformer, MW molecular weight, $R_s$ hydrodynamic radius, and $N$ number of trajectory points.

The operational principle of NSM is based on dark-field light-scattering microscopy imaging of a single nanochannel illuminated by supercontinuum visible light. The total scattering intensity recorded from a region of a nanochannel of length $L = 3\pi/2k$, where $k$ is the wave vector, with a biomolecule inside, can be approximated as[20]

$$I_t \approx I_c + I_m - \sqrt{2I_c I_m}. \tag{1}$$

Here, $I_t$ is the total scattered intensity from the nanochannel with a molecule inside, $I_c$ the scattered intensity from the nanochannel filled with solution but without a molecule, and $I_m$ the scattered intensity from the molecule alone[20]. Hence, by subtracting the empty nanochannel image from the same channel's image with a molecule, we acquire a differential dark-field image containing only $-\sqrt{2I_c I_m} + I_m$ (Fig. 1C). $I_m$ can be neglected because it is orders of magnitude smaller than the interference term $-\sqrt{2I_c I_m}$ since the scattering intensity of subwavelength objects scales with volume squared and the volume of the nanochannel is much larger than the volume of the molecule. Hence, the interference term generates the optical contrast—the key NSM feature enabling direct imaging of diffusing biomolecules and their oligomers, as demonstrated for multiple proteins, including Bovine Serum Albumin (BSA), in our seminal NSM work[20]. From a kymograph, that is, a time sequence of differential images acquired in this way, the MW and $R_s$ of each individual object can be extracted from its integrated optical contrast (iOC) and diffusivity ($D$), respectively[20] (see SI Section Molecular Property Characterization). As a second key point, we highlight the importance of the nanochannel cross-section area, $A$, for the iOC, and thus the LoD of NSM. For illustration, we designed an experiment using two nanochannels

$A_{III} = 153\,\text{nm} \times 32\,\text{nm} (4900\,\text{nm}^2)$ and $A_{IV} = 118\,\text{nm} \times 30\,\text{nm} (3540\,\text{nm}^2)$ to image BSA, which at 66 kDa MW is very close to the reported LoD of NSM[20] (Fig. 1D). As evident from the corresponding kymographs obtained using standard data preprocessing (see Methods), the BSA trajectory is only distinctly visible in the smaller channel $A_{IV}$ since iOC is inversely proportional to $A$[20]. This corroborates the potential of smaller nanochannels to reduce the LoD of NSM.

## Hierarchical vision transformer (h-ViT)

To enable characterization of ultrasmall molecules, smaller nanochannels alone are not enough. Thus, we develop a deep learning architecture, h-ViT, that builds on the Vision Transformer concept[37] (Fig. 2 and SI Section Deep Learning Architecture) and corresponding recent advances in attention mechanisms and transformer-based architectures[38] to effectively capture the multi-scale nature of the kymographs produced by NSM. Specifically, it estimates the desired molecular properties (MW, $R_s$, and number of trajectory points, $N$)—or, more generally, particle properties—without explicitly recreating said particles' pixel-wise kymograph trajectories. h-ViT does so by generating probability maps that encode the likelihood of particle presence at a specific position in the nanochannel at a specific point in time. This has the distinct conceptual advantage that a global attention mechanism can be used to focus on relevant features of the kymographs, which reduces the impact of local noise and enhances predicted data interpretability. This is important because in data regimes with very low Signal-to-Noise Ratio (SNR), accurately reconstructing the pixel-wise trajectories of particles becomes impossible due to the overwhelming noise that obscures individual particle movements[39]. Compared to previously reported H-Vit models[40], our model operates

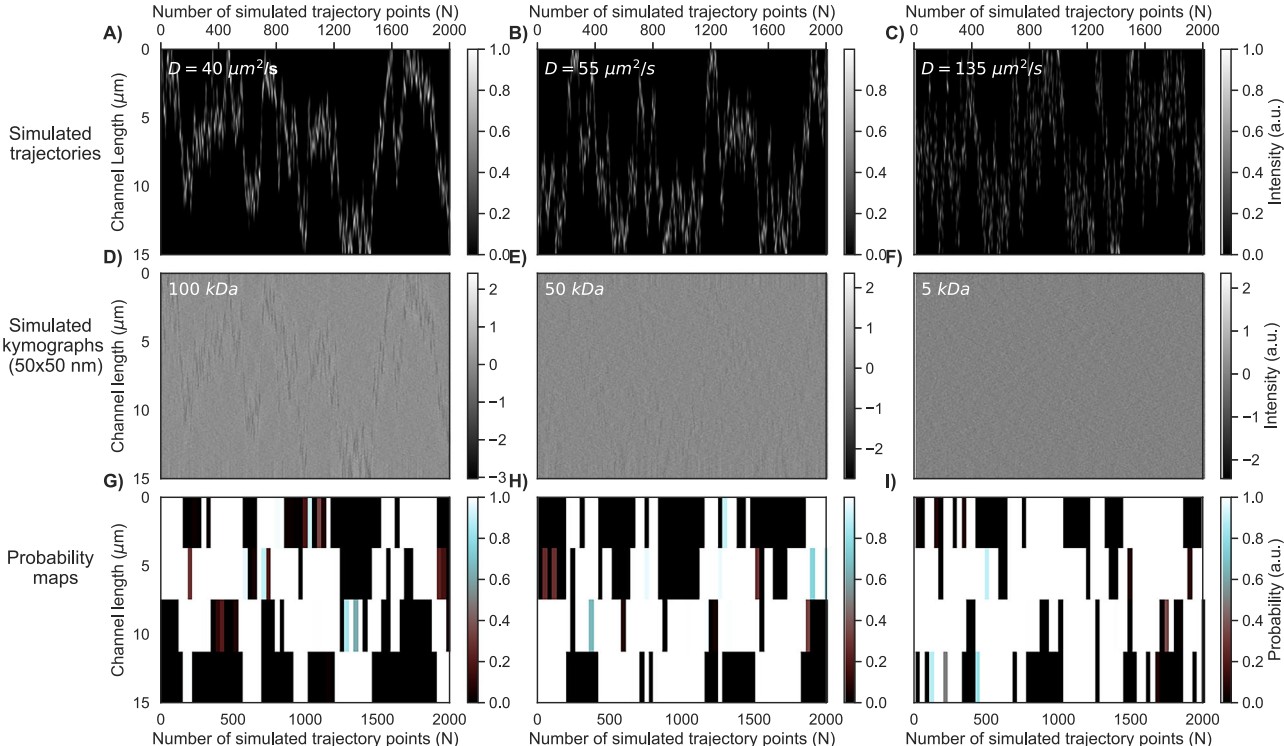

**Fig. 3 | Analysis of simulated single molecule trajectories with the h-ViT.** Diffusive trajectories for single particles simulated within a length that corresponds to the field of view of the camera in an NSM experiment and depicted as function of number of simulated trajectory points, *N*, with diffusivities: **A** *D* = 40 μm²/s; **B** *D* = 55 μm²/s; and **C** *D* = 135 μm²/s, respectively. Corresponding simulated example kymographs representing the light scattering signal from simulated 100 kDa, **D** 50 kDa, **E**, and 5 kDa, **F** particles, respectively, diffusing in a 50 × 50 nm² channel with diffusivities used in **A**–**C** combined with simulated background noise. Probability maps generated by h-ViT using the simulated kymographs as input, highlighting regions of low and high likelihood of particle presence for the 100 kDa, **G** 50 kDa, **H** and 5 kDa, **I** particles. As the key point, the probability maps represent areas of high particle presence probability identified by h-ViT even when no particle trajectory is obviously resolved in the input kymograph. We note that the coarse resolution along the nanochannel direction is the consequence of h-ViT looking at 7 scales of downsampling of kymographs that are 512 pixels long (i.e., 512/2⁷ = 4). For visualization purposes in this figure, the parts of the kymographs in which the trajectory is out of view of the camera are stitched out. Here, NSM denotes nanofluidic scattering microscopy, h-ViT hierarchical Vision Transformer, *N* number of simulated trajectory points, and *D* diffusivity.

on kymographs, jointly outputs probability and property maps, and integrates them over space-time rather than predicting displacement fields. Given the different objectives, direct benchmarking is not meaningful.

To explain the probability map generation by h-ViT, we use simulated data (see SI section Simulations of Particle Trajectories). Specifically, we simulated particle diffusion trajectories within a channel length that corresponds to the typical field of view of the microscope used for the NSM experiments further below, for three different *D* (Fig. 3A–C). Subsequently, these trajectories are used to construct simulated kymographs by modeling the optical response of each particle based on its light scattering properties in an arbitrarily chosen small nanochannel with a cross-section 50 × 50 nm², and by combining this signal with simulated background noise. As expected, the scattering signal from particles smaller than 100 kDa drops below the noise (Fig. 3D). Yet, inputting these simulated kymographs to h-ViT, downsampled by a factor of 2⁷, yields probability maps even when no trajectory is resolved (Fig. 3G–I).

### Assessing h-ViTs fundamental limits of MW and $R_s$ prediction

To assess the fundamental limits of h-ViT in terms of MW and $R_s$ predictions, we trained it on simulated kymographs with experimentally relevant nanochannel dimensions to generate probability maps from the ground truth trajectories. Specifically, we simulated 24 kymographs (SI Section Simulated Dataset) in the smallest below experimentally used channel $A_{II}$ = 63 × 30 nm² for each predicted MW and $R_s$ point in the range 1–30 kDa (Fig. 4A–C)

and 1–2.7 nm (Fig. 4D–F). The predicted values are weighted by the probability map such that, in regions of high probability, the corresponding MW and $R_s$ values contribute more to the final prediction. We furthermore categorize the obtained data according to the number of simulated trajectory points used (*N* = 100, 2000, and 10,000), since *N* mediates the amount of data available to the model, which is critical for reducing noise. We find that the relative mean error (μ, 1- the prediction accuracy) and 1- the relative standard deviation (σ, the prediction precision) for both MW and $R_s$ predictions increase with *N*, where *N* = 10,000 yields close agreement between true and predicted MW (Fig. 4C) and $R_s$ (Fig. 4F) for particles as small as 5 kDa.

It is now instructive to plot MW and $R_s$ prediction accuracy and precision values vs. *N* (Fig. 4G–J) for a simulated particle with MW = 6 kDa again in a nanochannel with $A_{II}$ = 63 nm × 30 nm to emulate the Insulin (MW = 5.8 kDa) experiment discussed below. Clearly, the accuracy in both MW (Fig. 4G) and $R_s$ (Fig. 4H) predictions increases for increasing *N* and approaches μ = 3.8% (MW) and μ = 3.7% ($R_s$), respectively, for *N* > 10,000. Importantly, for the prediction precisions of MW (Fig. 4I) and $R_s$ (Fig. 4J), we find very low σ-values for short trajectories (*N* < 2500), a maximum at *N* ≈ 6200, and asymptotically decreasing values for *N* > 6200, as the consequence of h-ViT being a biased estimator, which impacts the regime of very small and intermediate *N* (SI Section h-ViT as a Biased Estimator).

To finally put the h-ViT MW and $R_s$ prediction precision for the 6 kDa particle as a function of *N* into a more fundamental perspective, we have calculated the corresponding theoretical limits set by the

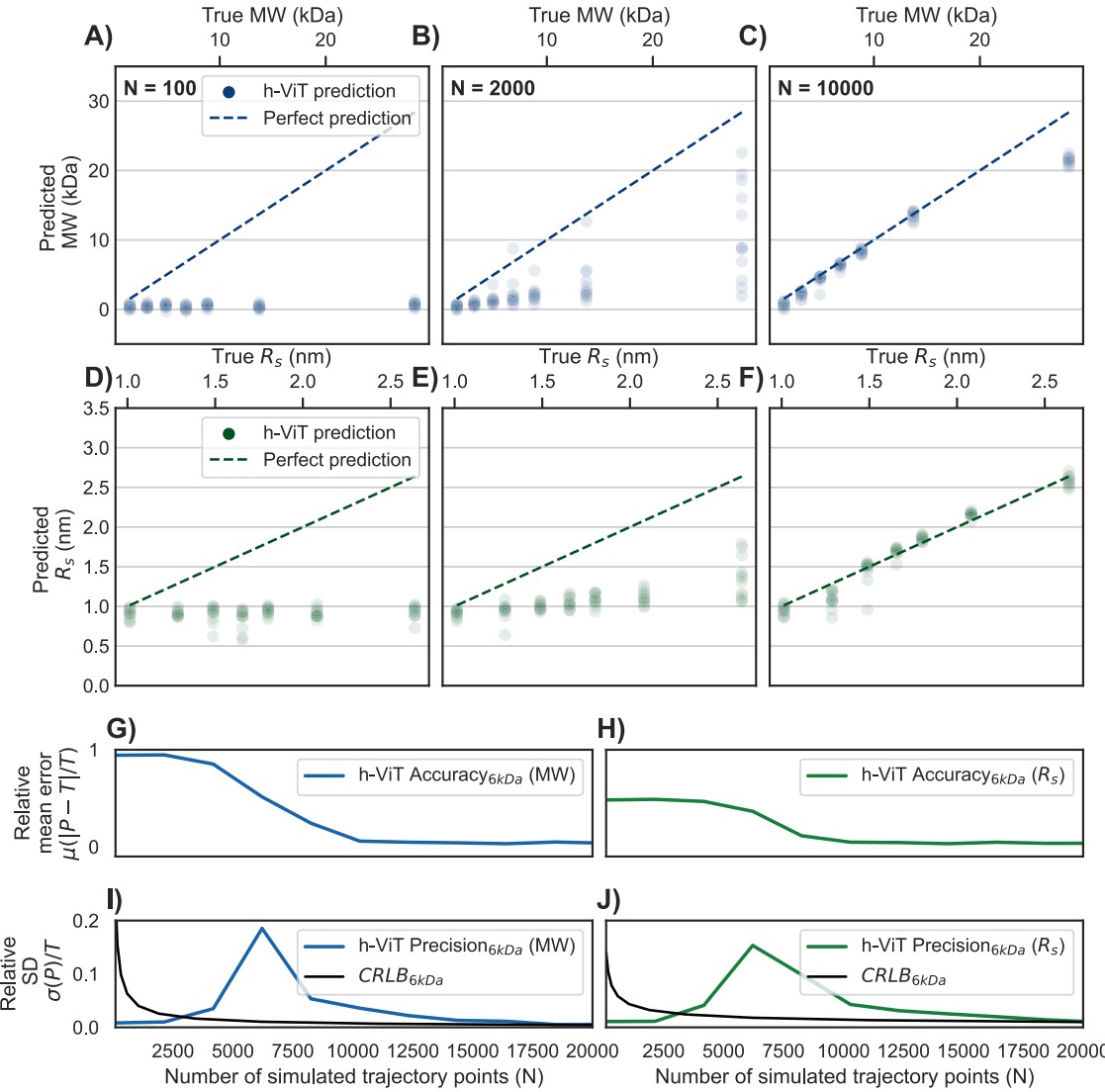

**Fig. 4 | Fundamental limits of h-ViT prediction.** Scatter plots of h-ViT-predicted MW vs. ground truth MW in the range of 1–30 kDa, based on simulated kymographs with increasing number of trajectory points: $N = 100$, panel A); $N = 2000$, panel B); and $N = 10,000$, panel C) for the smallest later experimentally used nanochannel cross section $A_{II} = 63 \times 30$ nm². Each blue dot represents a simulated particle with a different MW. The blue dashed line represents the ideal prediction line (perfect MW match). Scatter plots of predicted $R_s$ vs. ground truth $R_s$ in the range of 1–2.7 nm, based on the same simulated kymographs as in (A–C), with increasing number of trajectory points: $N = 100$, **D** $N = 2000$, **E** and $N = 10,000$, (**F**). Each green dot represents a simulated particle with different $R_s$. The green dashed line represents the ideal prediction line (perfect $R_s$ match). Relative mean error $\mu$ (1-accuracy) of h-ViT-predicted MW, (**G**) and $R_s$, (**H**) values for a 6 kDa particle. Relative standard deviation $\sigma$ (1-precision) of h-ViT-predicted MW,

(**I**) and $R_s$, (**J**) values for a 6 kDa particle as a function of the number of simulated trajectory points $N$. The black lines correspond to the Cramér–Rao Lower Bound (CRLB) that constitutes the theoretical limit of the highest possible prediction precision of MW, (**I**) and $R_s$, (**J**) by the h-ViT model. Notably, this fundamental limit is approached for large $N > 10,000$. The weak reduction in h-ViT performance observed for the largest MWs arises from edge-of-support under-representation during training that renders the regressor's bias a dominating effect, rather than true SNR limits. No experimental control group applies in this simulation benchmark, because predictions are evaluated directly against known ground-truth parameters. See Supplementary section Simulated Dataset for details. Here, h-ViT denotes hierarchical Vision Transformer, MW molecular weight, $R_s$ hydrodynamic radius, $N$ number of simulated trajectory points, CRLB Cramér–Rao lower bound, and SNR signal-to-noise ratio.

Cramér–Rao Lower Bound (CRLB) (SI Section Cramér–Rao Lower Bound). Evidently, for large $N$, the model performs near the CRLB-limit (Fig. 4I, J). It also shows how h-ViT makes predictions with higher precision than the CRLB in the regime where $N$ is small, due to its systematic tendency to predict MWs close to zero and $R_s$ close to 1 nm when the signal becomes indistinguishable from noise (SI Section h-ViT as a Biased Estimator). At very small N, the estimator is biased toward a conservative null estimate, which transiently reduces the empirical spread. As N grows, the bias relaxes, producing a shallow maximum, and precision then approaches the CRLB. This CRLB analysis illustrates how h-ViT makes predictions that align closely with the

true particle properties, even for a particle as small as 6 kDa, when $N > 10,000$.

## DNA ladder NSM measurements with h-ViT data analysis

To put h-ViT to an initial test on experimental NSM data, we selected a double-stranded DNA (dsDNA) ladder with 200, 100, and 50 bp molecules. Given the larger channel cross-section purposely used here ($A_I = 122 \times 97$ nm²), compared to our seminal NSM study[20] where we also measured 200 bp dsDNA in an $A = 110 \times 72$ nm² channel, the iOC of the 200 bp fragments measured here in channel $A_I$ is equivalent to roughly half the iOC of BSA measured in our seminal work.

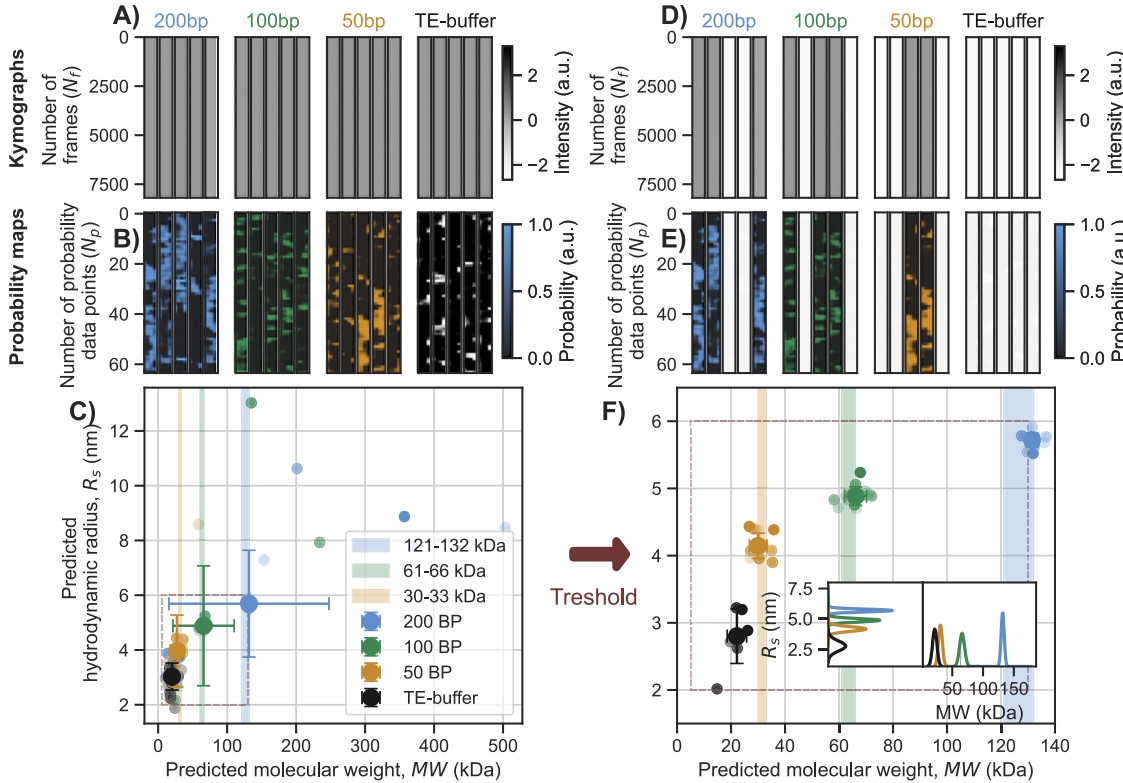

**Fig. 5 | DNA ladder characterization with NSM and h-ViT. A** Experimentally measured NSM kymographs for 200, 100, and 50 bp double-stranded (ds) DNA fragments in TE buffer, and for a pure TE-buffer control, respectively serving as input to the h-ViT model (see SI section Complete DNA Measurements for larger versions of the kymographs). Notably, no molecule trajectories are resolved in the kymographs, since all considered MWs are below the LoD of NSM for the used nanochannel size in combination with only the standard data treatment.
**B** Probability maps generated by the h-ViT model using the kymographs in (**A**) as input. They indicate regions of high and low likelihood of DNA molecule presence in the nanochannel and are color-coded by DNA molecule type (color key shown in the figure). We again note that the coarse resolution along the nanochannel direction is the consequence of h-ViT looking at 7 scales of downsampling of kymographs that are 512 pixels long (i.e., $512/2^7 = 4$). **C** h-ViT-model predicted $R_s$ values plotted versus the corresponding predicted MW values for the three different dsDNA fragment types, together with the empty channel pure TE-buffer control. Points represent individual measurements ($n = 20$ kymographs per dsDNA fragment type) and their opacity reflects the number of trajectory points, $N$, with high molecule presence probability available to h-ViT to make the prediction (more opaque points correspond to larger $N$, ranging from $N = 637$ to $N = 9128$). The error bars depict the standard deviations in $R_s$ and MW from the mean. The color-shaded

areas denote the nominal MW ranges (shaded backgrounds) obtained, i.e., 121 or 132 kDa (200 bp), 61 or 66 kDa (110 bp), 30 or 33 kDa (50 bp), when using the two different available calculation methods from the literature and supplier, respectively (see SI section dsDNA Length and Weight Calculation in Methods)[34,35].
**D–F** Same as (**A–C**), but after applying a threshold to the probability maps to improve detection accuracy, as outlined in Post-processing in "Methods". The threshold applied is defined by summing the total probability across each probability map, using the average probability sum over the only TE-buffer filled nanochannel control experiment to create the threshold value, and excluding completely any probability maps (and thus kymographs) that contain less than half the average probability sum of the TE-buffer control experiment. Inset histograms in (**F**) show the distributions of predicted MW and $R_s$ values after application of the threshold. The error bars depict the standard deviations in $R_s$ and MW from the mean after application of the threshold. For the full measurement series and corresponding results, see SI Section Complete DNA Measurements. Here, NSM denotes nanofluidic scattering microscopy, h-ViT hierarchical Vision Transformer, bp base pairs, dsDNA double-stranded DNA, TE Tris--EDTA buffer, MW molecular weight, $R_s$ hydrodynamic radius, LoD limit of detection, $N$ number of trajectory points, and SD standard deviation.

Consequently, in channel $A_I$, all three DNA ladder samples are purposely below the LoD of NSM in its reported form[20]. The raw kymographs (Fig. 5A and Supplementary Figs. 1–3, panels a–f) were preprocessed as outlined in the preprocessing section in "Methods".

Inputting these experimental kymographs to h-ViT produces probability maps that highlight regions of high dsDNA presence likelihood for all three dsDNA fragment sizes (Fig. 5B). Based on these probability maps, h-ViT predicts MW mean values from 20 kymographs as MW = 142.6 ± 116.1, MW = 68.5 ± 45.6 kDa, and MW = 29.1 ± 8.8 kDa, respectively. For the two largest dsDNAs, these numbers are slightly outside the range of nominal values, 121/132 kDa, defined by the used calculation method[34,35] ('dsDNA Size and Weight Calculation' in "Methods"). Turning to the predicted $R_s$, we note that dsDNA behaves like stiff rods rather than spherical particles, which means that their $D$ is distinctly lower at a corresponding MW compared to, e.g., a globular protein[41]. Accordingly, the predicted mean

$R_s$ = 5.7 ± 1.9, $R_s$ = 4.9 ± 2.1, $R_s$ = 4.0 ± 1.3 nm are lower than the true fragment lengths 17, 34, and 68 nm, respectively, and thus reflect the molecular shape (see dsDNA Length and Weight Calculation in "Methods").

Returning to the predicted MW values, agreement with nominal values is not perfect (SI Section Full DNA Measurements). Figure 5D–F report medians, the probability-sum threshold removes probability maps that most likely correspond to the nanochannel only being filled with buffer, narrowing dispersion without shifting the center. Here, we recall that h-ViT-predicted molecular properties from each kymograph are weighted by the probability map. Hence, we can introduce a probability threshold below which a kymograph is excluded from further analysis (Post-Processing and Thresholding in Methods). Replotting with this threshold applied, we find the effective elimination of $R_s$ and MW predictions that deviated from the median (Fig. 5D–F). This, in turn, leads to tighter distributions, i.e.,

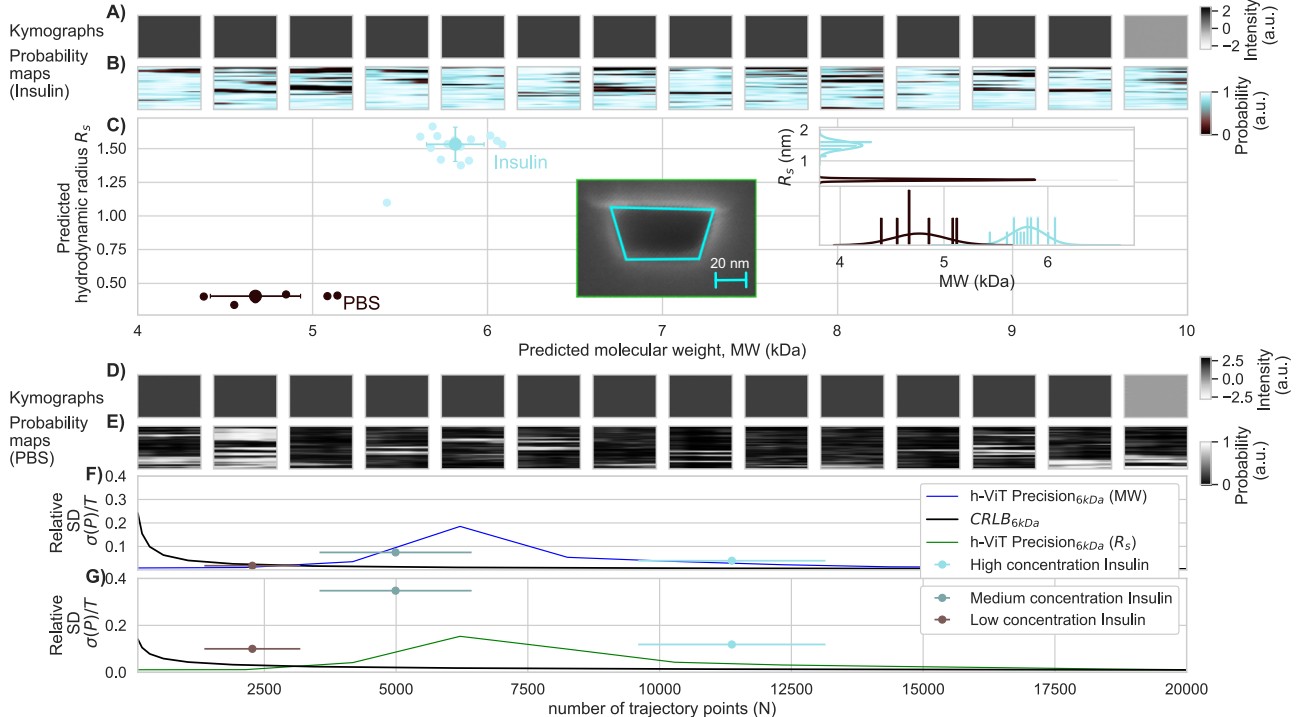

**Fig. 6 | Insulin molecule characterization. A** Representative kymographs from Insulin characterization experiments with NSM, serving as input to the h-ViT model. No molecule trajectories can be resolved in the kymographs, since all considered MWs are below the LoD of NSM in combination with only the standard data treatment, even for the used smallest nanochannel size. **B** Corresponding probability maps generated by h-ViT, indicating regions of high Insulin localization probability inside the nanochannel. **C** h-ViT-predicted $R_s$ versus MW values. Each data point corresponds to one kymograph (Insulin: $n = 20$ kymographs; PBS buffer control: $n = 100$ kymographs). Where shown, error bars indicate mean ± standard deviation across kymographs after applying a probability threshold, defined by summing the total probability across each probability map, using the average probability sum over the only PBS-buffer filled nanochannel control experiment to create the threshold value and excluding completely any probability maps (and thus kymographs) that contain less than half the average probability sum of the PBS-buffer control experiment. Individual points represent individual measurements (kymographs) and their opacity reflects the number of trajectory points, $N$, with high molecule presence probability available to h-ViT to make the prediction (more opaque points correspond to larger $N$, up to $N = 13,446$ for Insulin). We note

that for a small and fast diffusing molecule like Insulin, this means that kymographs with large enough $N$ in most cases only are produced if multiple molecules diffused through the nanochannel while the kymograph was experimentally obtained. Also shown are correspondingly obtained data points for pure PBS buffer control measurements, as extracted from kymographs in (**D**) and probability maps in (**E**). **F** Relative standard deviation $\sigma$ (precision) of h-ViT MW prediction as function of number of trajectory points $N$ based on simulated trajectories of length between 100 and 20,480 trajectory points for a 6 kDa molecule (blue line; simulation) plotted together with experimental Insulin measurements (points) at low ($N = 2272$), medium ($N = 4990$) and high concentration ($N = 11,370$) and the Cramér--Rao Lower Bound (CRLB) that illustrates the theoretical limit of prediction. Here, $\sigma$ is computed across predicted values using the mean as the measure of center. **G** Same as (**F**) but for $R_s$. Here, NSM denotes nanofluidic scattering microscopy, h-ViT hierarchical Vision Transformer, PBS phosphate-buffered saline, MW molecular weight, $R_s$ hydrodynamic radius, LoD limit of detection, $N$ number of trajectory points. $\sigma$ relative standard deviation (precision), and CRLB Cramér–Rao lower bound.

$R_s = 5.7 \pm 0.11$ nm, $R_s = 4.9 \pm 0.14$ nm, $R_s = 4.1 \pm 0.19$, and mean MWs now in very good agreement with the larger of the two nominal values[35], i.e., MW $= 131.8 \pm 2.5$ kDa (nominal: 132 kDa), MW $= 65.9 \pm 4.0$ kDa (66 kDa), MW $= 30.7 \pm 3.2$ kDa (33 kDa). We also note that the threshold indeed reduces the number of apparent particles detected in the TE-buffer negative control, leading to a clearer separation from the 50 bp dsDNA population (SI section h-ViT as a Biased Estimator). Finally, we note that whilst the trajectories of 200 and 100 bp dsDNAs can clearly be identified frame-by-frame and separated into individual molecules, the trajectories of 50 bp DNA molecules are invisible (SI Section Full DNA Measurements). This suggests that even in regimes of very low signal, accurate characterization of molecular properties is possible.

## h-ViT-enabled NSM characterization of 5.8 kDa Insulin molecules

To demonstrate h-ViT on a second and more challenging set of experimental NSM data, we nanofabricated ultrasmall nanochannels for NSM, $A_{II} = 63 \times 30$ nm², to investigate the peptide hormone Insulin in 1× PBS buffer at 3.2 μM concentration. This concentration was chosen to ensure an average number of Insulin trajectory points in

each measured kymograph, $N$, in the regime >95% precision and accuracy defined by the CRLB, i.e., $N > 10,000$ (Fig. 4G–J). This means that likely several individual Insulin molecules were moving through a nanochannel during the acquisition of a kymograph due to their high diffusivity. Hence, they together contributed to the h-ViT MW, $R_s$, and $N$ predictions per kymograph, which, thus, in most cases, constitute an average from several single Insulin molecules. We applied the same probability threshold strategy introduced above (Post-processing section in "Methods" and SI Section Full Insulin Measurements for data without probability threshold). As the nanochannel cross-section shrinks, the correspondingly increasing confinement reduces D due to increasing collisions with the nanochannel walls. To account for this in our analysis, we apply a nanochannel-specific hindered-diffusion calibration for the $R_s$ conversion, preventing systematic bias across nanochannel geometries. For further details on this calibration, see ref. 20.

Insulin has a nominal MW $= 5.8$ kDa and $R_s \approx 1.5$ nm[42]. As expected for this ultrasmall MW, even kymographs measured for both Insulin and PBS-buffer control in this ultrasmall nanochannel (comprised of $\approx 11,370$ frames equaling a maximum $N \approx 11,370$) do not resolve any

trajectories (Fig. 6A, D). Nevertheless, inputting the kymographs to h-ViT produces probability maps that highlight regions of high Insulin localization probability (Fig. 6B). It also finds a small number of maps for the PBS-control with probabilities above the threshold (Fig. 6D). Subsequently, based on these probability maps and the set threshold, h-ViT predicts the MW and $R_s$ values for each measured kymograph for Insulin and PBS-control (Fig. 6C). As input for the prediction h-ViT requires the nanochannel cross-section area, which we derived from post-mortem cross-section SEM images by taking the mean value of trapezoids fitted to three independent images of the same channel to account for the uncertainty in this estimation that stems from the limited resolution of the SEM due to charging effects during imaging (inset Fig. 6C and SI Section Nanochannel Area Estimation). This yields $A_{II} = 1891$ nm². Using this value, we find a population with median MW = 5.95 kDa $\pm$ 0.16 kDa and $R_s$ = 1.52 nm $\pm$ 0.13 nm, which is close to the Insulin literature values of 5.8 kDa and 1.53 nm, respectively[42]. When it comes to the population detected in the PBS control, it is clear that it is distinctly shifted to smaller $R_s$ and MW and thus likely caused by a combination of molecule-like noise patterns and possibly real tiny particles. The reason that it is absent in the Insulin measurements is that Insulin presence makes it unlikely that it will affect the scattering signal above the threshold.

Finally, it is useful to compare these results with the theoretical limits obtained above for a 6 kDa simulated particle in a channel with identical cross-section (c.f. Fig. 4). However, to acquire comparable experimental data would require many Insulin kymographs with (roughly) the same number of systematically varied trajectory points $N$. While this is practically impossible, to emulate it, we executed two additional experiments using Insulin at lower concentrations, i.e., 10 and 5 μg/ml, to effectively vary the expected average number of trajectory points, $N_{avg}$, for a given number of frames constituting the kymographs, i.e., again ≈11,370 frames (Supplementary Section Full Insulin Measurements). Subsequently, we used these datasets to generate the corresponding precision values, $\sigma$, of the MW (Fig. 6F) and $R_s$ (Fig. 6G) predictions at the three different concentrations (and thus different $N_{avg}$) by comparing the spread in the predicted MW and $R_s$ values against their respective means, normalized by the total number of trajectory points $N$ in the kymographs. This enables a rough quantification of how the prediction precision in the data increases with increasing $N$, also in the experiment. Comparing these values to the trends obtained for the entirely simulated 6 kDa particle (c.f. Fig. 4I, J) reveals that a maximum is reproduced qualitatively reasonably well in the experiment for both MW and $R_s$ (Fig. 6F, G). Furthermore, comparing the experimental points to the CRLB obtained above for the simulated 6 kDa particle, which represents the highest theoretically obtainable prediction precision for MW and $R_s$ given the specific noise characteristics of the experimental system at hand, reveals that provided a large enough $N$ (i.e., number of frames measured for a single molecule trajectory in an NSM experiment), the precision in MW and $R_s$ prediction likely approaches the CRLB also in the experiment. This, in turn, means that NSM measurements in the sub-10 kDa regime, in principle, appear possible with precisions very close to the theoretical limit, provided kymographs with large enough $N$ can be produced.

## Discussion

We have recently introduced NSM, which enables label-free detection and characterization of MW and $R_s$ of freely diffusing biomolecules down to an approximate LoD of 60 kDa[20]. The focus of this work has been to push this LoD, and the limit of MW and $R_s$ characterization, to below 6 kDa, by applying ultrasmall nanofluidic channels and developing the deep learning architecture h-ViT, which estimates MW and $R_s$ without explicitly recreating the pixel-wise NSM single molecule trajectories. Applying h-ViT first to a nanochannel $A_I = 122 \times 97$ nm², we successfully detected and characterized MW and $R_s$ of a dsDNA ladder

comprised of 200, 100, and 50 bp molecules. Subsequently, using an ultrasmall nanofluidic channel $A_{II} = 30 \times 63$ nm², we detected and characterized the peptide hormone Insulin to 5.95 kDa MW $\pm$ 0.16 kDa and 1.52 nm $R_s$ ($\pm$0.13 nm), respectively, thereby lowering the LoD of NSM by a factor of 10. Finally, we benchmarked these results by comparing them to the theoretical limit set by the CRLB and found that this fundamental limit is within reach, provided the number of trajectory points per kymograph is high enough, i.e., $N > 10,000$, for the given noise characteristics of our current experimental setup. This highlights a corresponding experimental challenge, since the intrinsically high $D$ of very small molecules reduces their time in the microscope field of view, which at a given frame rate results in smaller $N$ for smaller particles. To overcome this limitation in the future and enable the characterization of even smaller molecules, as well as truly single-molecule trajectories also in this low SNR regime, we suggest the use of, e.g., statistical bootstrapping[43], electrostatic entrapment[44], or entropic traps[45]. While the characterization of oligomers and, consequently, molecular mixtures is possible with demonstrated standard NSM workflows[20], in the sub-10 kDa regime in focus here, we do not reconstruct single-molecule trajectories. As a consequence, mixture quantification likely requires orthogonal priors or upstream fractionation to be possible. Taken together, this supports NSM as a label-free microscopy approach for biologically relevant families of molecules in the sub-10 kDa size regime, such as cytokines, chemokines, and peptide hormones.

Looking forward, we envision the use of NSM in general for label-free determination of homogeneity, oligomeric state, and stoichiometry of biological molecular samples or for investigations of molecular interactions in solution. The specific concepts developed in this work enable such studies even in biologically highly relevant families of molecules in the sub-10 kDa size regime, which have remained inaccessible to label-free single-molecule microscopy due to their small sizes. Furthermore, as already demonstrated[20], NSM can also be applied to characterize biological nanoparticles in terms of size and optical contrast, which in turn is linked to their refractive index and, as we predict, to their molecular cargo loading, such as, for example, mRNA. As a further forward-looking aspect, we note that specificity beyond molecular size and weight may be obtained either by introducing surface modifications inside nanofluidic channels by adapting surface chemistry developed for open surfaces[46] that enables tailored specific binding of single molecules to engineered receptors or to explore the possibility to introduce orthogonal selectivity by geometric size exclusion/filtering or nanofluidic fractionation[47–49] in combination with NSM readout on the same chip.

## Methods

### Sample preparation

50, 100, and 200 bp dsDNA fragments (NoLimits from ThermoFisher) were used in the experiments and diluted to a range of 12–30 nM in 1× TE buffer (Calbiochem from ThermoFisher).

Bovine serum albumin - BSA (UltraPure from FisherScientific) was diluted to 100 nM in 1× PBS buffer (MP Biomedicals from FisherScientific).

Insulin (SigmaAldrich) was diluted to a range of 0.8–3.2 μM in 1× PBS buffer (MP Biomedicals from FisherScientific).

### dsDNA length and weight calculation

The length of double-stranded DNA (dsDNA) fragments was calculated using the structural parameters of B-form DNA, which is the predominant form under physiological conditions. For B-form DNA, each base pair contributes approximately 0.34 nm to the overall length of the molecule[50]. To determine the length of a dsDNA fragment, we thus multiplied the number of base pairs (bp) by 0.34 nm to obtain 17, 34, and 68 nm lengths for the 50, 100, and 200 bp dsDNA fragments, respectively.

The MW of a dsDNA fragment can be estimated in two ways: (i) using the widely accepted approximation of 660 g/mol per base pair found in the literature[35]. This value accounts for the average molecular weight of the four nucleotides (A, T, G, and C) along with the sugar-phosphate backbone. As an example, for a 200 bp dsDNA fragment, MW is calculated as:

$$200 \times 660 = 132,000 \, \text{g/mol} \, (132 \, \text{kDa}). \tag{2}$$

While this approximation is generally accurate for most applications, the exact MW may vary slightly depending on the specific base composition, as G-C pairs are slightly heavier than A-T pairs[35]. Hence, there is an intrinsic uncertainty in the calculation of MW for DNA molecules and no perfectly accurate ground truth to compare with. Furthermore, the supplier of the dsDNA molecules that we used, ThermoFisher Scientific, advertises a slightly different way of calculating MW, generally delivering slightly smaller values[34]. Since it is not within the scope of this work to argue for a preferred method to calculate the MW of DNA molecules, we included both estimates in our work and thus provide a range of nominal MW values rather than a distinct number.

Because the dn/dc of dsDNA ($\approx 0.172 \, \text{mLg}^{-1}$) differs from the canonical protein value by <10%, we used the same conversion constant ($a = 0.46 \, \text{Å}^3\text{Da}^{-1}$) for all analytes; the resulting $\approx 8\%$ bias is smaller than the experimental scatter and does not alter the conclusions.

## Nanofluidic chips

The nanofluidic chips used contain a series of nanochannels with tailored cross-sectional dimensions as indicated in the main text and 80 μm length, which are connected to macroscopic in- and outlets through two microchannels with cross-section dimensions of $50 \times 1.5 \, \mu\text{m}^2$. The transport of the 30 μl liquid sample from the inlets to the nanochannels is controlled by pressurizing the inlets to 2 bar. During the measurements, the applied pressure is turned off to solely rely on diffusion for molecular motion through the nanochannels during imaging.

Fabrication of the nanofluidic chips was carried out in cleanroom facilities of Federal Standard 209 E Class 10-100, using electron-beam lithography (JBX-9300FS/JEOL Ltd), photolithography (MA6/Suss MicroTec), reactive-ion etching (Plasmalab 100 ICP180/Oxford Plasma Technology and STS ICP and PlasmaTherm/Advanced Vacuum), magnetron sputtering (MS150/FHR), deep reactive-ion etching (STS ICP/STS) and wet oxidation (wet oxidation/centrotherm), fusion bonding (AWF 12/65/Lenton), scanning electron microscopy (Supra 55VP/Zeiss) and dicing (DAD3350/Disco). In particular, the fabrication comprised the following processing steps of a 4-inch silicon (p-type) wafer of 1 mm thickness:

Thermal oxidation: (1) cleaning for 10 min at 80 °C in 1:1:5 H2O2:NH3OH:H2O (SC-1), rinsing in water, HF-dip for 30 s, cleaning for 10 min at 80 °C in 1:1:5 H2O2:HCl:H2O (SC-2), rinse in water, and drying under N2-stream. (2) Wet oxidation in a water atmosphere for 750 min at 1050 °C (2100 nm thermal oxide).

Fabrication of alignment marks: (1) spin coating HMDS adhesion promoter (MicroChem) at 3000 rpm for 30 s and soft baking on a hotplate (HP) at 115 °C for 120 s. Spin coating S1813 (Shipley) at 3000 rpm for 30 s and soft baking (HP) at 110 °C for 1 min. (2) Expose alignment marks for 12 s in the contact aligner at 6 mW cm-2 intensity. (3) Development in MF-319 (Microposit) for 60 s, rinsing in water, and drying under N2-stream. (4) Reactive-ion etching (RIE, PlasmaTherm) for 15 s at 250 mTorr chamber pressure, 50 W RF-power and 80 sccm O2-flow (descum). RIE for 45 min at 100 mTorr chamber pressure, 100 W RF-power, and 40 sccm CF4-flow (900 nm etch depth in thermal oxide). (5) Removal of resist in 50 ml H2O2 + 100 ml H2SO4 at 130 °C for 10 min, rinsing in water and drying under N2-stream.

Fabrication of nanochannels: (1) Spin coating 950k-PMMA A2 (MicroChem Corporation) at 2000 rpm for 60 s and soft baking (HP) at

180 °C for 5 min. (2) Electron-beam exposure at 1 nA with a shot pitch of 2 nm and 2600 μC cm-2 exposure dose. (3) Development in IPA 93 : 7 H2O for 60 s at 6 °C and drying under N2-streaI. (4) RIE (Plasmalab) for 5 s at 40 mTorr chamber pressure, 40 W RF-power, and 40 sccm NF3-flow. (5) RIE (Plasmalab) for 60 s at 20 mTorr chamber pressure, 100 W RF-power, 30 sccm CF4-flow, and 20 sccm CHF3-flow. (6) Removal of the PMMA mask in 50 ml H2O2 + 100 ml H2SO4 at 130 °C for 10 min, rinsing in water, wet-etching in standard Cr-wet etch, and drying under N2-stream.

Fabrication of microchannels: (1) spin coating HMDS at 3000 rpm for 30 s and soft baking (HP) at 115 °C for 2 min. Spin coating S1813 (Shipley) at 3000 rpm for 30 s and soft baking (HP) at 110 °C for 1 min. (2) Expose microchannels for 12 s in a contact aligner at 6 mW cm-2 intensity. (3) Development in MF-319 (Microposit) for 60 s, rinsing in water, and drying under N2-stream. (4) RIE for 15 s at 60 mTorr chamber pressure, 60 W RF-power, and 60 sccm O2-flow (descum). RIE for 20 min at 30 mTorr chamber pressure, 275 W RF-power, 50 sccm Ar-flow, and 50 sccm CHF3-flow (1500 nm etch depth in thermal oxide). (5) Removal of resist in 50 ml H2O2 + 100 ml H2SO4 at 130 °C for 10 min, rinsing in water and drying under N2-stream.

Fabrication of inlets (from backside): (1) magnetron sputtering of 500 nm Al (hard mask). (2) Spin coating S1813 at 3000 rpm for 30 s and soft baking (HP) at 110 °C for 1 min. (3) Expose inlets for 12 s in the contact aligner at 6 mW cm-2 intensity. (4) Development in MF-319 for 60 s, rinsing in water and drying under N2-streaI. (5) Aluminum wet etch (4:4:1:1 H3PO4: CH3COOH: HNO3: H2O) for 10 min to clear the hard mask at inlet positions. (6) RIE for 30 min at 30 mTorr chamber pressure, 275 W RF-power, 50 sccm Ar-flow, 50 sccm CHF3-flow. (7) Deep RIE for 1500 cycles of 12 s at 5 mTorr chamber pressure, 600 W RF-power, 10 W platen power, 130 sccm SF6-flow (Si-etch), and 7 s at 5 mTorr chamber pressure, 600 W RF-power, 10 W platen power, and 85 sccm C4F8-flow (passivation). (8) Removal of Al-hard mask in 50 ml H2O2 + 100 ml H2SO4 at 130 °C for 10 min, rinsing in water and drying under N2-stream.

Fusion bonding: (1) cleaning of the substrate together with a lid (175 μm thick 4-inch pyrex, UniversityWafers) in 5:1:1 H2O:H2O2:NH3OH (SC-1) for 10 min at 80 °C. (2) Prebonding the lid to the substrate by bringing surfaces together and applying pressure manually. (c) Fusion bonding of the lid to the substrate for 5 h in N2 atmosphere at 550 °C (5 °C min-1 ramp rate).

Dicing of bonded wafers: cutting nanofluidic chips from the bonded wafer using a resin-bonded diamond blade of 250 μm thickness (Dicing Blade Technology) at 25 krpm and 2 mm s-1 feed rate.

Once the nanofabrication processing parameters are established for a specific nanofluidic channel size range, the nanofabrication yield is high and independent of channel cross-section. Prior to use, defective chips are screened out by routine optical inspection and flow testing before use. For measurements, we define success at the kymograph level as meeting a probability-sum threshold (via the H-ViT) sufficient for property estimation, with additional rejection of stationary hotspots in the kymograph indicative of surface adsorption and of recordings showing unstable background. The principal failure modes of NSM experiments are nanochannel clogging, molecular unspecific adsorption, and insufficient $N$ when molecules diffuse out of the field of view before adequate integration. In our typical NSM workflow, we mitigate these unwanted effects by inline filtration, buffer optimization, channel preconditioning flow, and operating within an empirically determined concentration window that avoids both molecular depletion and crowding. When introducing the smallest nanochannels used in this work, initial trials with these systems exhibited lower experiment success rates. However, once the experimental protocol was tuned, the pass fraction of experiments is comparable and high across all used nanochannel sizes. In other words, smaller nanochannels do not intrinsically reduce the rate of successful measurements.

## Experimental NSM setup

The NSM optical setup is a darkfield microscope. A commercial microscope platform (RM21, Mad City Labs) is used with a micro-positioner for sample positioning and a nanopositioner for fine alignment, both by Mad City Labs. The nanofluidic chip (see Section Nanofluidic Chips) is placed in a custom-made chip holder to interface with a pressure controller (Fluigent MFCS-EX). Pressure is applied independently to four sample solution reservoirs to control flow and exchange sample solutions. Illumination is provided by a super-continuum laser (NKT SuperK FIU-15), filtered to 500–800 nm via a variable spectral filter (NKT SuperK Varia) with a maximum power up to 550 mW after filtering. The illumination light is focused at the back focal plane of the objective (Nikon 60×, NA 1.49). Two small right-angle mirrors guide light in and out of the imaging path. The illumination light produces a spot of about 10 μm (HWHM), but is not imaged. Scattered light is imaged by a sCMOS camera (Zyla 5.5), where each pixel corresponds to 27.4 nm on the sample. The region of interest was 600 × 30 pixels, acquired at approximately 6000 fps with on-chip 30-frame accumulations, resulting in a recorded frame rate of 200 fps.

Reference-frame subtraction acts as a high-pass filter on possible slow light source fluctuations, and at our illumination conditions, the dominant noise terms are photon shot noise and camera read noise[20]. Consequently, the LoD is effectively source-agnostic in the experimental configuration used in this work.

## Simulations of particle trajectories

For small biomolecules, such as the proteins and dsDNA fragments imaged in this work inside a nanofluidic channel, their scattering profile in our system is effectively that of a diffraction-limited spot. We therefore simulate their response as point scatterers, corresponding to the optical response of biomolecules with four main varying properties: integrated optical contrast iOC, diffusivity $D$, Gaussian width $s$ and velocity $v$. At time zero, a position $x_0$ is randomly chosen along the nanochannel, and the position $x_i$ of the molecule at frame $i$ is generated as Brownian motion with $x_i = x_{i-1} + v\Delta t + \mathcal{N}(0,1)\sqrt{2D\Delta t}$, where a random value $\mathcal{N}(0,1)$ is drawn from a normal distribution with mean 0 and standard deviation 1. The optical response of the biomolecule was then simulated along this path as a Gaussian of width $s$, positions $x_0 \ldots x_k$ for the length of trajectory $k$, and magnitude iOC.

The generated response was then combined with simulated background noise (Extended Data Fig 3B) as $I = I^0 \cdot I^r$, where $I^0$ is the response of the empty channel and $I^r$ the response of the biomolecule, and kymographs were created according to the procedure described in "Methods" section "Preprocessing".

The simulated noise is generated as described in the following. We assume that the noise in the system is dominated by two sources: the shot noise from the channel, imperfections and dirt in the channel, and the vibration of the channels. The shot noise is generated as an inevitable consequence of quantum-level fluctuations in the number of photons interacting with the sensors of the camera over short periods of time. Shot noise, as a sequence of independently occurring events of photon interaction with a constant mean rate, can be modeled through a Poisson distribution, which in turn can be modeled by a Gaussian distribution in the limit of large samples. The second source of noise is the vibrations of the channel, modeled as a harmonic vibration in the centroid of the Gaussian distribution. Thus, for each frame $t$, the background is first initialized as

$$b_0 = e^{(-(x-(x_0+x_\lambda))^2/\lambda^2)} \cdot (1 + \mathcal{N}(0,1)^*b) \cdot (1 + A\mathcal{N}(0,1)) \quad (3)$$

where $x$ denotes the position along the channel, $x_0, \lambda, A$ are numerical parameters randomly drawn from a normal and uniform distributions,

respectively, * denotes the convolution operation, and where

$$x_\lambda = (2\mathcal{N}(0,1) + \sin((\pi - 0.05)t)) \cdot d_x,$$
$$b = e^{(-(x)^2/C^2)} + x_\lambda / \sum(e^{(-(x)^2/C^2)} + x_\lambda),$$

where $d_x$, $C$ are numerical parameters randomly drawn from uniform distributions. Here, in equation (3), the first factor is the aforementioned (partially sinusoidally variant and partially randomly sampled) Gaussian. The second factor corresponds to noise resultant of a perfect channel (1) plus noise resultant of dirt and imperfections in the channel $\mathcal{N}(0,1)^*b$, which can be modeled by Gaussian noise with a large correlation length. We achieve this by a convolution with a generic (randomly sampled) point spread function of the microscope and the (randomly sampled) position and size of said dirt. The third factor corresponds to the aforementioned shot noise and is effectively a randomly sampled scalar.

Finally, the total background noise $b_f$ is calculated as

$$b_f = b_0 \cdot (1 + N_\theta \mathcal{N}(0,1)) + .4N_\theta \mathcal{N}(0,1) \quad (4)$$

where a new normal variable is drawn for each position $x$ along the channel, and $N_\theta$ is a numerical parameter drawn from a uniform distribution correlated with the total quantitative noise level in the system.

The values of the numerical parameters are chosen such that they roughly correspond to the quantitative levels of noise seen in experimentally measured noise of our NSM microscope, within a numerical span large enough such that any quantitative variations in the noise are still appropriately accounted for. In principle, so long as the true values of the experimentally measured noise are encapsulated within the range of the simulated noise, we can expect h-ViT to properly interpolate and correct for any such variation of noise. However, note that this approach is sensitive to sources of noise which are not included in the simulations. Specifically, if a qualitatively different source of noise excluded in the simulations is present in a measurement at inference time (i.e., spurious vibrations as a result of human movement, or device malfunction), it will be ambiguous to h-ViT whether this portion of the data should be considered signal or noise. This problem can, to an extent, be mitigated by transfer-learning the trained network on a few examples of simulated trajectories within experimentally measured noise before each new set of measurements, effectively constituting a re-calibration step of the network. Alternatively, it is always possible to include new sources of noise into the simulations. However, note that this increases the state-space of possibilities which the network needs to learn, and thereby increases the overall training time and increases the requirements for representational ability within the choice of network architecture. In this work, we therefore use simulations of particle trajectories superimposed on experimentally measured scattering of an empty channel. The model is train-validated every time the loss converges (and the simulated MW, $R_s$ values are updated, see SI Section Curriculum Learning) against 250 simulated kymographs (of size 512 × 512, 512 × 1024, 512 × 2048, 512 × 4096, 5128192) with experimentally measured channel noise, using an 80–20 train-validation split. The channels that the noise is measured from are $A_I$ for the DNA measurements and $A_{II}$ for the Insulin measurements.

To preclude overfitting, h-ViT was trained exclusively on procedurally simulated kymographs that are re-generated on-the-fly each epoch with new trajectories, particle properties (MW, $R_s$), channel geometries, and noise measured from empty channels; no experimental frames were used for training or model selection. Convergence of training was monitored only on held-out simulated data. Generalization was then assessed solely on experimental datasets not seen during training: a dsDNA ladder in $A_I$ = 122 × 97 nm² and insulin in $A_{II}$ = 63 × 30 nm², where the model finds literature-consistent

properties (insulin: 5.95 ± 0.16 kDa; $R_s$ = 1.52 ± 0.13 nm) and cleanly separates buffer-only controls after probability-thresholding. Moreover, prediction precision versus trajectory length follows the Cramér–Rao lower bound, indicating performance limited by measurement statistics rather than memorization.

In practice, per-kymograph normalization and the probability-weighted property head make the model tolerant to moderate illumination or camera changes, so no retraining is required in those cases. By contrast, material changes in channel cross-section or temperature alter the $D \to R_s$ and iOC → MW mappings. Our default rule is therefore to (i) regenerate matched synthetic data and retrain end-to-end, or (ii) freeze the probability head and re-calibrate the property head with a short geometry-specific calibration. We adopt option (ii) when the imaging statistics are stable and only the physical mapping shifts, and option (i) when noise statistics and geometry both change.

### Preprocessing

The raw NSM microscope image data were pre-processed to transform them into kymographs by the following process. First, the intensity of the raw CMOS image data was normalized according to

$$\bar{I}(x,t) = \frac{I(x,t) - \langle I(x,t) \rangle}{\langle I(x,t) \rangle} \tag{5}$$

represents the time average of said intensity. Second, a low-pass-filtered version of $\bar{I}(x,t)$ was calculated by using two normalized sliding windows of sizes 200 × 1 and 1 × 200, and subtracted from $\bar{I}(x,t)$.

### Post-processing and thresholding

To improve the accuracy and precision of MW and $R_s$ predictions, we employed a two-step thresholding process on the probability maps generated by the h-ViT model. This post-processing process filters out low-confidence kymographs and thereby removes outliers that could distort the overall predictions. The first step involves filtering out kymographs where the likelihood of detecting true molecular signals is low, i.e., those dominated by noise. To achieve this, we calculated the total sum of probabilities across each probability map:

$$P_{\text{total}} = \sum_{x,t} P(x,t) \tag{6}$$

where $P(x, t)$ is the probability at position $x$ and time $t$ in the probability map. We then set a threshold $P_{\text{thresh}}$ based on control experiments (e.g., buffer-only measurements) by computing the average probability sum $\langle P_{\text{control}} \rangle$ for the control samples. The threshold applied in this work was then accordingly set to half of this control average:

$$P_{\text{thresh}} = \frac{1}{2} \langle P_{\text{control}} \rangle \tag{7}$$

We excluded any kymograph where $P_{\text{total}} < P_{\text{thresh}}$, ensuring that only high-confidence probability maps-likely corresponding to actual molecules-were retained for further analysis.

After applying the probability threshold, we refined the MW and $R_s$ predictions by identifying and removing outliers. For both the MW and $R_s$ values obtained from the retained kymographs, we fitted Gaussian distributions to the data. For each parameter (e.g., MW), we estimated the mean ($\mu$) and standard deviation ($\sigma$) of the Gaussian distribution:

$$f_{\text{Gaussian}}(\text{MW}) = \frac{1}{\sigma\sqrt{2\pi}} \exp\left( -\frac{(\text{MW} - \mu)^2}{2\sigma^2} \right) \tag{8}$$

Outliers were identified as points falling outside three standard deviations ($3\sigma$) from the mean, i.e., data points where:

$$\text{MW} > \mu + 3\sigma \text{ or } \text{MW} < \mu - 3\sigma \tag{9}$$

The same process was applied to the $R_s$ values, using the corresponding Gaussian distribution:

$$f_{\text{Gaussian}}(R_s) = \frac{1}{\sigma\sqrt{2\pi}} \exp\left( -\frac{(R_s - \mu)^2}{2\sigma^2} \right) \tag{10}$$

Any data points for MW or $R_s$ falling outside three standard deviations were removed from the final dataset.

By combining these two thresholding steps-first, by filtering based on probability and second by removing outliers using Gaussian fitting, we improved the precision and accuracy of the predicted MW and $R_s$ values. The thresholding step effectively excluded low-confidence kymographs, while the Gaussian fitting reduced the influence of noise-driven outliers, as reflected in the tighter clustering of data points and reduced variance in the scatter plots.

**h-ViT model and training.** The h-ViT architecture begins with the input processing stage, where the kymograph-a two-dimensional representation of scattered light intensity over time-is standardized by normalizing it with respect to its temporal standard deviation. This pre-processing step ensures that the input data is consistent, facilitating the subsequent feature extraction process. The kymograph then undergoes an initial convolution through a layer with a kernel size of 7, utilizing leaky ReLU activation to preserve nuanced variations in the data. This is followed by a series of convolutional blocks, each comprising multiple convolutional layers that progressively increase the number of filters. These blocks serve to downsample the kymograph while simultaneously capturing essential features at varying spatial resolutions.

A key innovation of this architecture lies in its hierarchical multi-scale approach. As the model progresses through each convolutional block, the spatial dimensions of the feature maps are reduced via max-pooling operations. At each scale, patches are extracted from the feature maps and encoded using a custom PatchEncoder layer. These encoded patches, now representing condensed and contextually enriched portions of the original kymograph, are fed into a series of transformer blocks. Each transformer block integrates a multi-head attention mechanism, which allows the model to focus selectively on different regions of the patches, and a multi-layer perceptron, which refines these focused representations. The attention mechanisms within the transformer blocks are pivotal, enabling the model to capture long-range dependencies and subtle variations in the kymograph that are critical for accurately estimating molecular properties such as $D$ and iOC. At each scale, the model generates predictions for molecular properties through dense layers that process the outputs of the transformer blocks. These predictions are complemented by probability maps, which quantify the likelihood that specific regions of the kymograph contain particle trajectories. These maps are then resized to match the original kymograph dimensions, ensuring that the model's focus aligns spatially with the input data.

The outputs from the different scales are subsequently concatenated to form a comprehensive representation of the kymograph. This representation undergoes further processing through additional dense layers, culminating in final predictions for the entire kymograph. In this final stage, a probability map and property predictions are

generated through a series of convolutional and dense layers. The final predictions are derived using a weighted average, where the weights are informed by the probability maps, ensuring that regions with higher probabilities of containing particle trajectories contribute more to the final output.

The networks were trained using the ADAM optimizer[51] with a learning rate of $1e-4$ on approximately $640,000$ simulated kymographs in the range $0 \leq MW \leq 30$ kDa and $0.7 \leq R_s \leq 3$ nm through curriculum learning as elaborated upon in the Curriculum Learning section in the Supplementary Information. The input during training is simulated images (kymographs) of size $512 \times 8192$, and the output is a single value at each scale of MW and $R_s$, as well as the probability map in the last scale. The model underwent validation every 400 epochs (equivalent to 1800 simulated kymographs) against 150 simulated kymographs that incorporated experimentally observed noise, following an 80–20 train-validation distribution.

## Reporting summary

Further information on research design is available in the Nature Portfolio Reporting Summary linked to this article.

## Data availability

The data that support the findings of this study are freely available from Zenodo[52] at: https://doi.org/10.5281/zenodo.17405422.

## Code availability

The code used for data preprocessing, analysis, model training (h-ViT), and figure generation is publicly available at https://gitlab.com/langhammerlab/nsm-hvit. The repository includes documentation and instructions to reproduce the analyses and figures presented in this manuscript.

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

## Acknowledgements

The authors acknowledge financial support from the Swedish Foundation for Strategic Research project FFL15-0087, the Swedish Research Council (VR) Consolidator Grant project 2018-00329, the Czech Science Foundation project 22–18203S, and the European Research Council (ERC) under the European Union's Horizon Europe research and innovation program (101043480/NACAREI). Part of this research has been executed at the Chalmers Nanofabrication Laboratory, MC2. The authors also acknowledge the computer cluster Alvis, through which some of the computations were enabled by resources provided by the Swedish National Infrastructure for Computing (SNIC), partially funded by the Swedish Research Council through grant agreement no. 2018-05973.

## Author contributions

H.K.M. and C.L. conceived the project. H.K.M., D.M., and G.V. designed and implemented the h-ViT model. B.Y. performed all single-molecule experiments with initial guidance from B.S., J.F., and D.A. produced the nanofluidic chips. H.K.M. and C.L. wrote the paper with input from all authors. C.L. provided the funding for the project.

## Funding

## Competing interests

D.A., J.F., B.S., and C.L. are co-founders of Envue Technologies AB that markets Nanofluidic Scattering Microscopy. The remaining authors declare no competing interests.
