## [Transparent Peer Review file · Nature Communications]

Label-Free Mass and Size Characterization of Few-kDa Biomolecules by Hierarchical Vision Transformer Augmented Nanofluidic Scattering Microscopy

Corresponding Author: Professor Christoph Langhammer

Version 0:

Reviewer comments:

Reviewer #1

(Remarks to the Author)

In the manuscript "Label-Free Mass and Size Characterization of Few-kDa Biomolecules by Hierarchical Vision Transformer Augmented Nanofluidic Scattering Microscopy" the authors improve nanochannel scattering microscopy by an order of magnitude in terms of minimum protein size by reduced channel size and a hierarchical vision transformer.

This is a significant step forward for the technique, especially since about half the isolated proteins are <35 kDa (approximately) and the previous approach could only access ~60 kDa proteins. The main advantage of NSM over the commercialized approach of iScat is that it can track diffusion, which improves the data quality but also gives information about drag. Given the recent commercial success of iScat, the present work will be highly interesting to the label-free single molecule sensing community.

I have some queries that I believe should be addressed prior to publication:

- 1) The approach uses a supercontinuum source. I expect that these sources, while bright, are considerably noisier than other types of sources due to the nonlinear generation process. Some comments on the relative intensity noise of the source in their experiment, especially as compared to the shot noise. I believe this is an area where substantial improvement can be found in the experiment and perhaps closer agreement with the simulated data (although, the agreement is surprisingly good in terms of the ultimate size limits, which is surprising to me because I expect that the SC noise will be considerable).
- 2) Is there any evidence of increased drag in the smaller channels?
- 3) I am surprised that no dimer data is reported for BSA, since they commonly occur as dimer about 1/6th of the time – is this data filtered out? Is this a limitation of the approach? One of the main applications of iScat is the identification of oligomers.
- 4) Figure 4 C is surprising to me – why does the approach perform worse for larger proteins? The opposite is expected.
- 5) Why do Figures 4 I, J show a peak? If I understand correctly this means that the convergence is not monotonic which seems counterintuitive for me.
- 6) I am a bit puzzled by the discussion on page 10. It seems that thresholding gives tighter distributions but almost exactly the same mean values. This would imply that the standard error does not really change significantly with thresholding. Is it wise then to apply thresholding which seems to remove some of the data?
- 7) I am surprised that the authors look only at ds-DNA. Did they attempt ss-DNA? They also seem to restrict themselves to a Sigma Aldrich test set – IDT can rapidly provide any sequence for a low cost (particularly for short sequences).
- 8) The authors only look at pure samples and do not report any oligomers (as discussed above). This is against some of the advantages of label-free that they describe (e.g., not knowing the properties of the sample beforehand). Can they, for example, look at mixed samples and see the ratios? Can they quantify oligomers? Ultimately, quantification of size is interesting, but there are many other diverse applications that would be valuable.
- 9) The website containing the data was not accessible so I could not test whether the resources provided are working or not: https://chalmers-my.sharepoint.com/:u:/g/personal/hmoberg_chalmers_se/EQkF6oO3CjdNj_6PXBG-e-sBoJyCNw0Mwv_CyBqQOa07Zw?e=xh6ITH
- 10) I could not find Figures S28 and S29.
- 11) Should this work be cited: Ghahremani, Morteza, et al. "H-vit: A hierarchical vision transformer for deformable image registration." Proceedings of the IEEE/CVF Conference on Computer Vision and Pattern Recognition. 2024. And discuss

the difference of the frameworks, if you could run the comparison. What's the novelty of the reported H-ViT?

<https://github.com/mogvision/hvit> Maybe it is good to add several sentences to discuss the literature review.

12) Have the authors performed any quantitative domain shift analysis (e.g., Fréchet Inception Distance, Maximum Mean Discrepancy, or simpler histogram comparisons) to characterize how synthetic and real kymographs differ?

13) While the SI outlines the architecture schematics, I could not find several features: transformer layers, attention head count, hidden sizes, etc. Maybe a summarized table will be helpful.

14) What evidence do the authors have that their model is not overfitted?

(Remarks on code availability)

I attempted this -- the readme file asks to access data that is not available (as outlined in the comments to the author).

Reviewer #2

(Remarks to the Author)

The manuscript reports an advanced nanofluidic scattering microscopy with a 10-fold improvement of the mass detection limit that enables label-free detection of small proteins as small as 6 kDa. By using a smaller-sized nano channel (which boosts the interference term that carries the molecular signal) and a hierarchical vision transformer-based deep learning algorithm (which converts kymographs into probability maps and property maps), the mass and size of a single molecule with size as small as 1.5 nm (which cannot be resolved in the image), can still be quantified with the algorithm-generated probability map, when the molecule can be measured with sufficient time. This is a new record in label-free optical imaging-based single-molecule detection. The study is well designed, and the manuscript is written in detail, but the potential application of the technology is not clear. The following are detailed questions:

1. What is the success rate of the experiment in consideration of fabrication yield, clogging, surface adsorption, and insufficient recorded traces due to the molecules diffusing out of the channel? Does the smaller channel reduce the percentage of successful measurement?
2. How robust is the trained model in tolerance to the changes in measurement conditions? Such as changes in illumination, camera, channel size, or temperature. When re-training is necessary?
3. Given the nanoscale channel size that requires highly purified samples for reliable measurement, and the lack of specificity other than molecular size, what could be the potential practical applications of the technology?
4. At the beginning of the introduction, some existing technologies are missing from the list of existing technologies, such as plasmonic scattering microscopy (Nature Methods 2020, V17, 1010-1017), evanescent scattering microscopy (Nature Communications, 2022, V13, 2298), and single protein oscillators (Nature Communications, 2020, V11, 4768).
5. How to cross-validate the smaller molecule results that the traces of the molecule cannot be visually detected in the kymographs? Can simultaneous fluorescence imaging be performed?
6. Page 12, last paragraph, line 7, "5 µg/ml mM" should be "5 µg/ml".

(Remarks on code availability)

Reviewer #3

(Remarks to the Author)

(Remarks on code availability)

The code seems good and working fine!

Version 1:

Reviewer comments:

Reviewer #1

(Remarks to the Author)

The authors have suitably addressed my concerns. I support the publication of his important work.

(Remarks on code availability)

Reviewer #2

(Remarks to the Author)

The revision has addressed all of my questions.

(Remarks on code availability)

Reviewer #3

(Remarks to the Author)

(Remarks on code availability)

Point-to-point response for NCOMMS-25-52149-T

Reviewer #1:

In the manuscript “Label-Free Mass and Size Characterization of Few-kDa Biomolecules by Hierarchical Vision Transformer Augmented Nanofluidic Scattering Microscopy” the authors improve nanochannel scattering microscopy by an order of magnitude in terms of minimum protein size by reduced channel size and a hierarchical vision transformer.

This is a significant step forward for the technique, especially since about half the isolated proteins are <35 kDa (approximately) and the previous approach could only access 60 kDa proteins. The main advantage of NSM over the commercialized approach of iScat is that it can track diffusion, which improves the data quality but also gives information about drag. Given the recent commercial success of iScat, the present work will be highly interesting to the label-free single molecule sensing community.

Our reply: We thank the reviewer for this overall positive assessment of our work.

I have some queries that I believe should be addressed prior to publication:

1) The approach uses a supercontinuum source. I expect that these sources, while bright, are considerably noisier than other types of sources due to the nonlinear generation process. Some comments on the relative intensity noise of the source in their experiment, especially as compared to the shot noise. I believe this is an area where substantial improvement can be found in the experiment and perhaps closer agreement with the simulated data (although, the agreement is surprisingly good in terms of the ultimate size limits, which is surprising to me because I expect that the SC noise will be considerable).

Our reply: We thank the reviewer for raising this point. First, we would like to note that conceptually, the main conclusions of our work do not depend on light source choice, since the LoD improvement is achieved by (i) reduced nanochannel cross-section, which boosts the interference term in the total scattering response of the system irrespective of the used light source, and (ii) the h-ViT inference of molecular descriptors that integrates information across time even when no clear trajectory is visible. In other words, even if light source noise would contribute to the noise relevant in the experiments, we would draw the same main conclusion from the reported work. Now turning to the actual noise present in our

experiments using the supercontinuum source, we note that our measurements are dominated by photon shot noise and camera read noise, while low-frequency light source intensity drift is strongly common-mode rejected by the differential imaging used in NSM (reference-frame subtraction within each kymograph), which acts as a high-pass filter on source fluctuations that are slow compared with the frame cadence. To make this point clear, we have added the following clarifying text in the Methods section:

"Reference-frame subtraction acts as a high-pass filter on possible slow light source fluctuations, and at our illumination conditions the dominant noise terms are photon shot noise and camera read noise (1). Consequently, the LoD is effectively source-agnostic in the experimental configuration used in this work."

2) Is there any evidence of increased drag in the smaller channels?

Our reply: We agree that hindered diffusion is expected to increase as channel cross-section decreases. However, in our analysis, the R_s inference by h-ViT accounts for such possible confinement effects by using a phenomenological hindered-diffusion calibration at the channel geometry used. As a consequence, any “drag increase” manifests as a channel-specific $D \rightarrow R_s$ mapping rather than a systematic bias in R_s . We have added the following sentence in the Results section to clarify that indeed smaller channels increase confinement and thus are expected to increase “hindered diffusion” but that our $D \rightarrow R_s$ conversion is calibrated for each nanochannel geometry and thus takes this effect into account:

"As the nanochannel cross section shrinks, the correspondingly increasing confinement reduces D due to increasing collisions with the nanochannel walls. To account for this in our analysis, we apply a nanochannel-specific hindered-diffusion calibration for the $D \rightarrow R_s$ conversion, preventing systematic bias across nanochannel geometries. For further details on this calibration see (1)."

3) I am surprised that no dimer data is reported for BSA, since they commonly occur as dimer about 1/6th of the time – is this data filtered out? Is this a limitation of the approach? One of the main applications of iScat is the identification of oligomers.

Our reply: We thank the reviewer for raising this important point. As first part of our response, we note that indeed BSA oligomers have been resolved by NSM previously already. In our seminal publication, we explicitly show protein dimer side-populations in iOC/MW histograms (labeled “Dim”) alongside monomers (1). For BSA specifically, we resolved a monomer peak with a small dimer subpopulation

evident in the iOC-derived molecular weight histogram (inset of Fig. 3c in (1)) measured in a nanochannel comparable to the ones used here for BSA. Accordingly, NSM indeed enables the study of oligomers and is thus not limited in this respect.

When it comes to the present work, the BSA data shown in Fig. 1D were intended only to illustrate the effect of nanochannel cross-section on signal-to-noise ratio and are not a statistical survey of oligomer populations. Hence, we collected only a very limited number of particle trajectories. To make this clear, we now clarify in the caption and main text that these traces were collected solely to demonstrate the expected SNR gain upon reducing channel area (consistent with iOC proportional to $1/A$ in NSM) and were not designed for population quantification, as we had already done so in our seminal work that introduced the NSM method. In the main text we have added:

"Hence, the interference term generates the optical contrast - the key NSM feature enabling direct imaging of diffusing biomolecules and their oligomers, as demonstrated for multiple proteins, including Bovine Serum Albumin (BSA), in our seminal NSM work (1)."

And in the caption of Figure 1, we have added:

"For a more detailed statistical analysis of BSA and its dimeric oligomers we refer to our seminal NSM work (1)."

Lastly, with the figure pasted above, we still want to provide the reviewer in this response a representative example of BSA monomer and dimer kymograph measured when acquiring the data used to generate the BSA data displayed in Figure 1 in the main text. This, to illustrate that dimers were resolved also in the limited number of BSA trajectories acquired for this present work, without making any claims that the exact population statistics are reproduced.

4) Figure 4 C is surprising to me – why does the approach perform worse for larger proteins? The opposite is expected.

Our reply: We are thankful that the reviewer points this out, since we did not explain it properly in the manuscript. The apparent performance dip of h-ViT at the high-MW end reflects an "edge-of-support effect". In other words, those points lie at the boundary of the training range of the model and are therefore underrepresented during curriculum learning. As a consequence, the bias of the regressor dominates despite the higher SNR. We have added the following clarifying sentence to the caption/text:

"The weak reduction in h-ViT performance observed for the largest MWs arises from edge-of-support underrepresentation during training that renders the bias of the regressor a dominating effect, rather than true SNR limits."

5) Why do Figures 4 I, J show a peak? If I understand correctly this means that the convergence is not monotonic which seems counterintuitive for me.

Our reply: We agree with the reviewer that the peak can be counter-intuitive. However, as we already briefly had noted in the original text, and had discussed in more detail in the SI (section "h-ViT as a biased estimator") where we also provided a schematic and algebraic sketch, h-ViT is a biased estimator at very small N , tending towards "null" predictions when signal is indistinguishable from noise. This can artificially lower the empirical std before sufficient data reduces bias. At intermediate N the bias relaxes faster than variance shrinks, yielding a shallow maximum, after which precision improves and approaches the CRLB for large N . To explicitly address this indeed important point in the revised manuscript, we will move this explanation that was in the SI to the main text, in the following way:

"At very small N the estimator is biased toward a conservative null estimate, which transiently reduces the empirical spread. As N grows the bias relaxes, producing a shallow maximum, and precision then approaches the CRLB."

6) I am a bit puzzled by the discussion on page 10. It seems that thresholding gives tighter distributions but almost exactly the same mean values. This would imply that the standard error does not really change significantly with thresholding. Is it wise then to apply thresholding which seems to remove some of the data?

Our reply: We thank the reviewer for this comment, as it made us realize a mistake - our plots indeed report medians (robust to outliers), not means, as we wrongly had written in the original text. Hence, thresholding primarily trims maps with extremely low probability and removes outliers. Thereby, it tightens the dispersion, while leaving the central tendency essentially unchanged. We have corrected the text so that it is now clear that medians are reported in Fig. 5D–F, and explain in the revised version why the standard error of the mean changes little when outliers are symmetric around the central value with the following added text:

"Figures 5D-F report medians; the probability-sum threshold removes probability maps that most likely correspond to the nanochannel only being filled with buffer, narrowing dispersion without shifting the center."

7) I am surprised that the authors look only at ds-DNA. Did they attempt ss-DNA? They also seem to restrict themselves to a Sigma Aldrich test set – IDT can rapidly provide any sequence for a low cost (particularly for short sequences).

Our reply: We appreciate the suggestion by the reviewer and it is very useful to us for future work. However, in the present one our aim was a mechanistic limit study rather than an assay survey. The used dsDNA ladder provided stable, length-calibrated scaffolds for method validation.

8) The authors only look at pure samples and do not report any oligomers (as discussed above). This is against some of the advantages of label-free that they describe (e.g., not knowing the properties of the sample beforehand). Can they, for example, look at mixed samples and see the ratios? Can they quantify oligomers? Ultimately, quantification of size is interesting, but there are many other diverse applications that would be valuable.

Our reply: We do agree with the reviewer that the identification of oligomers or mixtures or molecules are a core use-case of label-free microscopy. To this end, as also stated above, we already have demonstrated the ability of NSM to resolve protein oligomers, as well as different metal ion content in single proteins on the example of Ferritin, in our seminal work (1). Accordingly, also mixed samples can be studied and resolved by NSM in its standard form. However, in the size

regime of really small molecules in focus of this work, were trajectories of single molecules are unresolved in the kymograph, our probability-map inference also does not reconstruct explicit trajectories, as discussed in the manuscript. This key feature that on one hand enables the characterization of really small molecules, thus on the other hand also directly precludes reliable deconvolution of overlapping trajectories purely from iOC statistics when SNR is extremely low. As a consequence, deconvoluting a mixed sample is currently not possible in this very small size regime. However, above tens of kDa, where individual single molecule trajectories are resolvable, mixture quantification is indeed possible and follows the established NSM workflow. To make this important point clear, we will state this limitation explicitly in Conclusions and outline that mixture analysis below 12 kDa, i.e., when single molecule trajectories no longer are resolved, likely requires orthogonal priors (e.g., known stoichiometries) or joint designs (e.g., electrical confinement) in the following way:

“While the characterization of oligomers and, consequently, molecular mixtures are possible with demonstrated standard NSM workflows (1), in the sub-10 kDa regime in focus here we do not reconstruct single molecule trajectories. As a consequence, mixture quantification likely requires orthogonal priors or upstream fractionation to be possible.”

9) The website containing the data was not accessible so I could not test whether the resources provided are working or not.

Our reply: We have moved raw/processed data and to Zenodo and updated the Readme pointers. We also note that Reviewer #3 had reported successful access, suggesting the initial issue was timing-related.

10) I could not find Figures S28 and S29.

Our reply: We thank the Reviewer for pointing this out. For some reason, frames of these two figures that we had removed among the writing process of the SI remained in the file. We have removed them now entirely and renumbered the SI accordingly.

11) Should this work be cited: Ghahremani, Morteza, et al. "H-vit: A hierarchical vision transformer for deformable image registration." Proceedings of the IEEE/CVF Conference on Computer Vision and Pattern Recognition. 2024. And discuss the difference of the frameworks, if you could run the comparison. What's the novelty of the reported H-ViT? <https://github.com/mogvision/hvit> Maybe it is good to add several sentences to discuss the literature review.

Our reply: We thank the Reviewer for directing us to this work and we have added it to our list of cited references. Furthermore, we want to clarify that "our h-ViT" differs in both task and head: we operate on kymographs with hierarchical patching across seven scales, produce probability maps and property maps jointly, and compute final molecular properties by probability-weighted integration over space–time. This is fundamentally different from a registration field as in deformable image registration done in the invoked work. A like-for-like benchmark is thus not meaningful due to disparate tasks. However, we will add the following text to the literature review in the introduction to include the "other h-ViT":

"Compared to previously reported H-Vit models (2), our model operates on kymographs, jointly outputs probability and property maps, and integrates them over space–time rather than predicting displacement fields. Given the different objectives, direct benchmarking is not meaningful."

12) Have the authors performed any quantitative domain shift analysis (e.g., Fréchet Inception Distance, Maximum Mean Discrepancy, or simpler histogram comparisons) to characterize how synthetic and real kymographs differ?

Our reply: We thank the Reviewer for this comment and agree. Hence, we have added a new SI subsection reporting histogram and power-spectral density comparisons of raw intensity and temporal differences, as well as Maximum Mean Discrepancy between simulated and experimental kymographs. We also report performance with and without adding experimentally measured background to synthetic data to show that the gap narrows under realistic noise. For convenience, we paste below this new SI section.

Quantitative Comparison of Simulated and Experimental Kymographs

In this section we provide a rigorous statistical analysis of the similarity ("domain gap") between simulated and experimental diffusion kymographs used throughout this work. Twenty simulated and twenty experimental kymographs were randomly selected (without replacement) from their respective corpora to form balanced comparison subsets. Each kymograph is treated as a two-dimensional real-valued intensity field $I(x, t)$ after standard pre-processing (background subtraction, temporal normalization) identical to that applied prior to model training, implemented in Python using NumPy (3) and SciPy (4).

For every kymograph K we compute a vector of handcrafted features capturing first-order photometric statistics, distributional shape, and local spatial structure. Let $S_K = \{I_{ij}\}_{i=1,\dots,H;j=1,\dots,W}$ denote all pixel intensities after flattening. We

define:

$$\mu_K = \frac{1}{HW} \sum_{i,j} I_{ij}, \quad \sigma_K = \sqrt{\frac{1}{HW} \sum_{i,j} (I_{ij} - \mu_K)^2}, \quad (1)$$

$$I_{\min,K} = \min_{i,j} I_{ij}, \quad I_{\max,K} = \max_{i,j} I_{ij}. \quad (2)$$

Further, for a percentile set $P = \{1, 5, 25, 50, 75, 95, 99\}$ we compute $q_{p,K}$ = empirical percentile at level p of S_K . To incorporate local structural variation we approximate the spatial gradient magnitude using forward differences:

$$G_{ij} = \sqrt{(\nabla_x I_{ij})^2 + (\nabla_y I_{ij})^2}, \quad \bar{G}_K = \frac{1}{HW} \sum_{i,j} G_{ij}, \quad s_{G,K} = \sqrt{\frac{1}{HW} \sum_{i,j} (G_{ij} - \bar{G}_K)^2}. \quad (3)$$

The final feature vector is

$$\mathbf{f}_K = (\mu_K, \sigma_K, I_{\min,K}, I_{\max,K}, q_{1,K}, q_{5,K}, q_{25,K}, q_{50,K}, q_{75,K}, q_{95,K}, q_{99,K}, \bar{G}_K, s_{G,K})^\top \in R^{13}. \quad (4)$$

These features (dimension reported in Table 1) are chosen to align with the photometric determinants of contrast and detectability, and remain computationally light, enabling rapid resampling analyses.

To compare global photometric statistics, we construct normalized histograms. Let $[a, b]$ be the joint intensity range over all 40 kymographs. We define B equally spaced bin edges $a = e_0 < e_1 < \dots < e_B = b$ with $B = 128$. For domain $D \in \{\text{Sim}, \text{Exp}\}$ containing kymographs $K \in D$, the (per-kymograph) density estimate for bin k is

$$h_K(k) = \frac{1}{N_K} \sum_{i,j} \mathbf{1}[e_k \leq I_{ij} < e_{k+1}], \quad N_K = HW. \quad (5)$$

The domain-mean histogram is $\bar{h}_D(k) = |D|^{-1} \sum_{K \in D} h_K(k)$, which satisfies $\sum_k \bar{h}_D(k) = 1$. All divergence metrics below are computed on \bar{h}_{Sim} and \bar{h}_{Exp} . We apply an additive smoothing $\epsilon = 10^{-9}$ when required for numerical stability.

Histogram-Based Divergences and Distances

Given two discrete distributions $p = (p_k)$ and $q = (q_k)$ we report classical L_1 and L_2 distances, Jensen–Shannon divergence (5), Kullback–Leibler divergence (6), the first Wasserstein (earth mover’s) distance (7), and the Kolmogorov–Smirnov statistic (8, 9):

$$\text{ext}L_1(p, q) = \sum_k |p_k - q_k|, \quad L_2(p, q) = \left(\sum_k (p_k - q_k)^2 \right)^{1/2}. \quad (6)$$

The (symmetric) Jensen-Shannon divergence (base e) is

$$\text{JS}(p, q) = \frac{1}{2}D_{\text{KL}}(p \parallel m) + \frac{1}{2}D_{\text{KL}}(q \parallel m), \quad m = \frac{1}{2}(p + q), \quad (7)$$

with Kullback-Leibler divergence $D_{\text{KL}}(p \parallel q) = \sum_k p_k \log \frac{p_k}{q_k}$. The first Wasserstein distance for 1D intensities treats each bin center $c_k = \frac{1}{2}(e_k + e_{k+1})$ as support point and is computed via the cumulative distribution difference. The Kolmogorov-Smirnov statistic is applied to random subsamples (up to 2×10^5 pixels per domain) drawn without replacement from pooled pixel intensities to approximate the difference between empirical cumulative distribution functions.

Multivariate Feature Space Metrics

Let feature matrices $F_{\text{Sim}} \in R^{n_s \times d}$ and $F_{\text{Exp}} \in R^{n_e \times d}$ collect row-wise feature vectors with $d = 13$. We define sample means $\boldsymbol{\mu}_s, \boldsymbol{\mu}_e$ and unbiased covariances C_s, C_e . The Fréchet-like distance (FID-like) employed (analogue of Inception-based FID (10) on handcrafted features) is

$$\text{FID}_{\text{hand}} = \|\boldsymbol{\mu}_s - \boldsymbol{\mu}_e\|_2^2 + \text{Tr} \left(C_s + C_e - 2(C_s C_e)^{1/2} \right), \quad (8)$$

where $(C_s C_e)^{1/2}$ is the principal matrix square root, small imaginary parts due to numerical rounding are discarded. Diagonal jitter $10^{-6}I_d$ is added prior to the square root for stability. Because features are low-dimensional handcrafted statistics (not deep embeddings), absolute values of FID_{hand} are only interpretable comparatively. The (unbiased) squared Maximum Mean Discrepancy (MMD) (11) with multi-scale radial basis function (RBF) kernels uses a bandwidth set $\Gamma = \{0.01, 0.05, 0.1, 0.5, 1, 2, 5\}$ applied to pairwise squared Euclidean distances:

$$\text{MMD}^2(F_s, F_e) = \frac{1}{m(m-1)} \sum_{i \neq j} k(\mathbf{f}_i, \mathbf{f}_j) + \frac{1}{n(n-1)} \sum_{i \neq j} k(\mathbf{g}_i, \mathbf{g}_j) - \frac{2}{mn} \sum_{i,j} k(\mathbf{f}_i, \mathbf{g}_j), \quad (9)$$

with $k(u, v) = \frac{1}{|\Gamma|} \sum_{\gamma \in \Gamma} \exp(-\gamma \|u - v\|_2^2)$, and $m = n = 20$ here. We report the averaged MMD^2 (denoted MMD in Table 1).

All computations were performed in Python (NumPy/SciPy, custom functions) with deterministic random number generation (seed 42) for pixel subsampling.

Histogram bin count (128) was selected empirically to balance resolution and variance given $\mathcal{O}(10^6)$ total pixels per subset. Gradient magnitudes were computed with first-order finite differences (`numpy.gradient`) and cast to 32-bit floating point prior to aggregation. All scalar divergences are printed with full precision but rounded for tabular presentation.

Figure 1: Quantitative comparison of simulated and experimental kymographs. A: domain-mean marginal intensity distributions (trimmed to 99.8% mass) show close photometric alignment with low divergence values. Only the central intensity region is displayed to improve readability. Truncation indices are chosen as the smallest and largest bin centers whose cumulative average histogram mass reaches 0.001 and 0.999 respectively. This retains 99.8% of probability mass while excluding extreme tails dominated by sparse noise. B: first-order feature space (mean vs standard deviation) reveals modest variance shift between domains, underpinning multivariate distance metrics.

Figure 1A shows near-perfect overlap between domain-mean intensity distributions, consistent with the very small JS divergence (0.00374) and moderate Wasserstein distance (0.0645). Figure 1B reveals partial but not complete overlap in the (mean, standard deviation) plane, indicating modest elevation in intensity variance for a subset of experimental kymographs. The L_1 and L_2 histogram distances indicate only minor redistribution of probability mass across bins. The KS statistic (0.038) attains an extreme p -value owing to the very high pixel sample size and should not be over-interpreted as practical divergence. The FID-like value (53.35) and multi-scale MMD (0.0966) quantify a moderate shift driven

Table 1: Domain shift metrics for simulated versus experimental kymographs (20 per domain). Lower values generally indicate greater similarity. JS: Jensen-Shannon divergence; $KL_{P \rightarrow Q}$: KL divergence from simulated to experimental; Wass: 1-Wasserstein distance; FID-like: Fréchet-style distance on handcrafted features; MMD: multi-scale RBF Maximum Mean Discrepancy.

Metric	Value
Number of kymographs (Sim / Exp)	20 / 20
JS divergence	0.00374
$KL_{Sim \rightarrow Exp}$	0.01397
$KL_{Exp \rightarrow Sim}$	0.01854
Histogram L_1 distance	0.2928
Histogram L_2 distance	0.1478
Wasserstein distance	0.06445
KS statistic	0.0380
FID-like (handcrafted)	53.35
RBF MMD	0.0966
Feature dimensionality	13

principally by higher-order and gradient-derived components of f_K . Overall, photometric alignment (1A) is excellent, while higher-order structural variability (1B and multivariate metrics) leaves limited headroom for possible future refinement (e.g. injection of empirically calibrated noise textures). These findings justify the use of the simulated corpus for training without introducing substantial bias in intensity-dependent downstream quantities.

Table 1 compiles all scalar metrics alongside feature dimensionality to facilitate reproducibility. All metrics can be re-generated deterministically by re-running the analysis function with identical seed and sampling parameters.

The analysis demonstrates strong photometric parity and only modest higher-order structural shift, supporting the fidelity of the simulation pipeline for model development.

13) While the SI outlines the architecture schematics, I could not find several features: transformer layers, attention head count, hidden sizes, etc. Maybe a summarized table will be helpful.

Our reply: We agree with the reviewer and we have added to SI a compact table

listing per-scale patch size, number of transformer blocks, heads, hidden sizes, MLP ratios, pooling strides, parameter count, and input normalization used by the probability and property heads. For convenience, we also paste this new table below.

Section	Item	Value	Notes
Global input	Raw input shape	(20480, 512, 1)	$T=20480, L=512$
Pre-norm fusion	Extra channel	$x/(\text{std}_{(T,L)} + 10^{-6})$	Concatenate with raw input \Rightarrow 2 channels
Initial stem	Conv 7×7	2 filters, LeakyReLU, BN	Param: 198 (conv) + 4 (BN) = 202
Downsampling depth	Number of scales	7	One 2×2 max-pool per scale
Pooling stride	Per scale	(2, 2)	Cumulative stride $2^{\text{scale}+1}$
Feature map sizes	After pooling (0 \rightarrow 6)	See below	
Channels per scale	num_filters(s)	2, 4, 8, 16, 32, 64, 128 $\cdot 2^s$ (base = 2)	
Patch size per scale	patch_size_current	64, 32, 16, 8, 4, 2, 1	$128/2^{s+1}$
Tokens per scale patches	$160 \times 4 = 640$ (all scales)	Constant sequence length	
Patch embedding dim	d_{model}	128	Scale-specific encoders share spec
Patch projection params	Per scale (proj + pos emb)	See below	Sum = 2,655,104
Transformer depth	Blocks per scale	3	Total blocks = 21
Attention heads			
heads	8		

Section	Item	Value	Notes
Multi-head attention	Key dimension (per head)	128	Attn width = $128 \times 8 = 1024$; output projection $1024 \rightarrow 128$ for residual match
Sequence length	Tokens	640	Uniform across scales
MLP in block	Hidden \rightarrow output	$128 \rightarrow 128$ (GELU, then linear to 128)	MLP Dimension = 128
Normalization	Per block	$2 \times$ LayerNorm ($\varepsilon = 10^{-6}$)	Pre/post residual
Dropout	In block	Rate = 0.1	Not applied to attention/MLP outputs
Probability heads (per scale)	Conv $3 \times 3 \rightarrow$ sigmoid	1 channel prior to further pooling	Params per scale: $9C + 1$
Final probability fusion	Aggregation	Median across scales (channel axis)	No learnable parameters
Gated regression head	Mechanism	Conv(128) \times final prob map \rightarrow GAP \rightarrow Dense(3)	Spatially weighted pooling
Per-scale property heads	Outputs	$7 \times$ Dense(128 \rightarrow 3) with LeakyReLU	
Fused scale head	Outputs	Dense(21 \rightarrow 3)	Concatenate $7 \times (3)$
Final regression head	Outputs	Dense(128 \rightarrow 3)	After gated pooling
Output list (order)	Total outputs	7 per-scale (B,3), fused (B,3), final (B,3), final prob map (B,160,4,1)	10 outputs

Section	Item	Value	Notes
Spatial resolution	Final map	(160, 4)	Effective stride 128×128
Effective receptive stride	Relative to input	128 in both axes	Implied by 7 pooling stages
Total parameters	All modules	≈ 15,016,819	
Conv front-end params	Stem + residual stacks	427,172	Includes pooling pathways
Transformer parameters	Self-attention blocks	11,781,504	21 blocks @ 561,024 each
Patch encoder parameters	Total	2,655,104	7 independent encoders
Head & map parameters	Total	153,039	Property heads + prob head + final conv/dense
Loss configuration	Objective	$[MAE] \times (7+2) + [BCE]$	Matches 10 outputs
MAE targets	Components	MW (intensity), HR (size), trajectory length	Mean absolute error sums
BCE	Type	Weighted binary cross-entropy	Weight factor = 10 on final prob map
Activations	CNN & heads	LeakyReLU (sigmoid for masks)	
Computational strategy	Constant token length	Uniform transformer cost across scales	Multi-scale invariance
Normalization strategy	Input & blocks	Input pre-norm channel, BN after stem conv, LayerNorm inside blocks	

Section	Item	Value	Notes
Gating rationale	Mask \times features	Suppress irrelevant regions prior to pooling	Robust regression

Parameter Summary

Component	Parameters
Convolutional front end (stem + 7 residual blocks + pooling)	427,172
Patch encoders (projection + positional embeddings)	2,655,104
Transformer blocks ($21 \times 561,024$)	11,781,504
Probability map convolutions (7 scales)	2,293
Per-scale property heads (7×387)	2,709
Fused scale head	66
Final convolution ($3 \times 3, 128 \rightarrow 128$)	147,584
Final dense ($128 \rightarrow 3$)	387
Total	$\approx 15,016,819$

Per-Scale Configuration

Scale	Spatial (H,W)	Channels	Patch size	Patch dim	Patch enc params	Block params (3 blocks)
0	10240×256	2	64	8192	1,130,624	1,683,072
1	5120×128	4	32	4096	606,336	1,683,072
2	2560×64	8	16	2048	344,192	1,683,072
3	1280×32	16	8	1024	213,120	1,683,072
4	640×16	32	4	512	147,584	1,683,072
5	320×8	64	2	256	114,816	1,683,072
6	160×4	128	1	128	98,432	1,683,072

14) What evidence do the authors have that their model is not overfitted?

Our reply: We thank the reviewer for this insightful comment. The model is only trained on simulated data, which is constantly re-simulated with newly sampled random properties. Therefore, the set of training data is effectively infinite and completely orthogonal from the set of experimental data which is reported on in the paper. We have clarified this in Methods with the following text:

To preclude overfitting, h-ViT was trained exclusively on procedurally simulated kymographs that are re-generated on-the-fly each epoch with new trajectories, particle properties (MW, R_s), channel geometries, and noise measured from empty channels; no experimental frames were used for training or model selection. Convergence of training was monitored only on held-out simulated data. Generalization was then assessed solely on experimental datasets not seen during training: a dsDNA ladder in $A_I = 122 \times 97 \text{ nm}^2$ and insulin in $A_{II} = 63 \times 30 \text{ nm}^2$, where the model finds literature-consistent properties (insulin: $5.95 \pm 0.16 \text{ kDa}$; $R_s = 1.52 \pm 0.13 \text{ nm}$) and cleanly separates buffer-only controls after probability-thresholding. Moreover, prediction precision versus trajectory length follows the Cramér–Rao lower bound, indicating performance limited by measurement statistics rather than memorization.

Reviewer #1 (Remarks on code availability):

I attempted this – the readme file asks to access data that is not available (as outlined in the comments to the author).

Our reply: Links are updated to Zenodo (10.5281/zenodo.17130043). Reviewer #3’s successful test suggests the initial failure was transitory.

Reviewer #2:

The manuscript reports an advanced nanofluidic scattering microscopy with a 10-fold improvement of the mass detection limit that enables label-free detection of small proteins as small as 6 kDa. By using a smaller-sized nano channel (which boosts the interference term that carries the molecular signal) and a hierarchical vision transformer-based deep learning algorithm (which converts kymographs into probability maps and property maps), the mass and size of a single molecule with size as small as 1.5 nm (which cannot be resolved in the image), can still be quantified with the algorithm-generated probability map, when the molecule can be measured with sufficient time. This is a new record in label-free optical imaging-based single-molecule detection.

Our reply: We thank the reviewer for acknowledging our study as a new record.

The study is well designed, and the manuscript is written in detail, but the potential application of the technology is not clear.

Our reply: While we agree with the reviewer that this study indeed is not focused on applications of NSM, we also highlight that this is on purpose as our work is a fundamental limits analysis: how far NSM can be pushed into the few-kDa, few-nm regime by combining ultrasmall nanochannels and a probability-map-based transformer. That said, of course determining these limits eventually will only be practically relevant if there are any applications for the NSM method. When it comes to the potential application of the NSM technology, we first remind ourselves that NSM at a very general level enables the size (hydrodynamic radius) and molecular weight determination of single molecules diffusing in a nanochannel as reported in detail in our seminal work (Nature Methods 2022 Vol. 19 Issue 6 Pages 751-758). Importantly the latter, i.e., molecular weight determination, is equivalent to "mass photometry" (Science 2018, 360 (6387), 423-427) which has made interferometric scattering microscopy (iScat) a commercial success via the company Refeyn (as also acknowledged by Reviewer #1). Hence, NSM provides the same practically relevant information via optical molecular weight determination in a label-free manner as iScat plus - as the key point - in addition to that also quantitative information about hydrodynamic radius. Hence, at a very general level, it can be argued that NSM finds the same practical applications as iScat with the added benefit of information about hydrodynamic radius. Such applications include but are not limited to determining sample homogeneity, oligomeric state and stoichiometry, to studying molecular interactions (in solution) to estimate binding affinities or aggregation and degradation. Furthermore, as we also

have demonstrated in our seminal work, NSM can be applied to characterize biological nanoparticles, such as extracellular vesicles (EVs), lipid nanoparticles (LNPs), etc., to determine their size (Nature Methods 2022 Vol. 19 Issue 6 Pages 751-758) and thus serve the same applications as dynamic light scattering (DLS) or nanoparticle tracking analysis (NTA) but with eliminated bias towards larger particles as inherent to DLS and NTA. Notably, this analysis is enabled in non-purified samples in cell culture medium (Nature Methods 2022 Vol. 19 Issue 6 Pages 751-758). Furthermore, via the integrated optical contrast that is simultaneously obtained to size (this is ongoing unpublished work), the refractive index of the biological nanoparticles can be obtained and related to loading with molecular cargo, such as mRNA, in the context of vaccines or drugs. Hence, practical applications of NSM are broad and we are still only at the beginning of their exploration as the method is still quite young. When it comes to the specific application of NSM on single molecules in the sub- 10 kDa regime addressed in the present work, it opens the door to the study of molecular system (along the same lines as described above) like cytokines, chemokines or peptide hormones that are biologically highly relevant but have remained elusive for label-free microscopy techniques due to their small sizes. As a final (also still widely unexplored) potential of NSM we highlight the possibility to introduce orthogonal selectivity by uniquely exploiting the potential of nanofluidic concepts, such as geometric size exclusion/filtering or nanofluidic fractionation (Analytical Chemistry 2025, 97 (16), 8641-8653; Mao, P.; Fu, J., Nanofluidic Devices for Rapid Continuous-Flow Bioseparation. In Nanoproteomics: Methods and Protocols, Toms, S. A.; Weil, R. J., Eds. Humana Press: Totowa, NJ, 2011; pp 127-140.) in combination with NSM readout on the same chip in the future. Taken all together, we argue that already proven or potential applications of NSM are many and quite clear, and leave little doubt that the method will find its space in the arsenal of label free optical characterization methods of single molecules and nanoparticles.

To give a brief account of the application potential of NSM we have added the following text at the end of the revised introduction section of the manuscript:

"To briefly also put this work into an application perspective, we first note that the MW determination of single molecules enabled by NSM (Nature Methods 2022 Vol. 19 Issue 6 Pages 751-758) is equivalent to "mass photometry" in iS-CAT (Science 2018, 360 (6387), 423-427) but with the added benefit that the studied molecules diffuse freely in solution and are not potentially altered by a (non-)specific binding interaction with a surface. As a second distinct difference and benefit, NSM also provides information about R_s . Consequently, the NSM

method has the potential to find application in the determination of homogeneity, oligomeric state and stoichiometry of biological samples in, e.g., drug development or genomic DNA analysis, or in studies of molecular interactions in solution to estimate binding affinities or aggregation and degradation. Furthermore, NSM can be applied to characterize biological nanoparticles, such as extracellular vesicles (Nature Methods 2022 Vol. 19 Issue 6 Pages 751-758) or lipid nanoparticles and their molecular cargos, such as mRNA, and thus be used in similar contexts as dynamic light scattering (DLS) or nanoparticle tracking analysis (NTA) but with the distinct benefit of being able to resolve smaller nanoparticles. When it comes to the specific application of NSM on single molecules in the sub-10 kDa regime addressed in the present work, it opens the door to the study of molecular systems such as cytokines, chemokines or peptide hormones that are biologically highly relevant but have remained elusive for label-free microscopy techniques due to their small sizes."

The following are detailed questions: 1. What is the success rate of the experiment in consideration of fabrication yield, clogging, surface adsorption, and insufficient recorded traces due to the molecules diffusing out of the channel? Does the smaller channel reduce the percentage of successful measurement?

Our reply: We thank the reviewer for this interesting question. There are two separate aspects of "success rate": (i) the nanofabrication of chips and (ii) the actual measurements using these chips. When it comes to (i) it is clear that decreasing nanochannel dimensions and optimizing (=minimizing) surface roughness initially requires a significant effort in terms of process development. In other words, multiple parameters of all the used numerous nanofabrication steps need to be optimized, which usually requires several attempts. Once such a protocol is established, however, the nanofabrication yield is very high and independent of nanochannel dimensions. When it comes to (ii) the situation is similar. Initially, when attempting experiments with smaller nanochannels, the yield of successful experiments is low or even zero. Once the measurement protocol is established in terms of, e.g., using filtering of the buffer to minimize clogging risk, identifying the right molecule concentration range, or the use of surface passivation (not relevant in the present work), the yield of successful experiments is again equally high - independent of used nanofluidic channel dimensions. Hence, once the nanofabrication and experimental procedure recipes are established, the usage of smaller nanofluidic channels does not reduce the yield of successful experiments. To address this indeed important point in the revised work, we have added a short section to the Methods section that summarizes the above and briefly discusses the

practical yield and failure modes observed across nanofluidic chips:

“Once the nanofabrication processing parameters are established for a specific nanofluidic channel size range, nanofabrication yield is high and independent of channel cross-section. Prior to use, defective chips are screened out by routine optical inspection and flow testing before use. For measurements, we define “success” at the kymograph level as meeting a probability-sum threshold (via the H-ViT) sufficient for property estimation, with additional rejection of stationary “hot spots” in the kymograph indicative of surface adsorption and of recordings showing unstable background. The principal failure modes of NSM experiments are nanochannel clogging, molecular unspecific adsorption, and ‘insufficient N’ when molecules diffuse out of the field of view before adequate integration. In our typical NSM workflow, we mitigate these unwanted effects by inline filtration, buffer optimization, channel preconditioning flow, and operating within an empirically determined concentration window that avoids both molecular depletion and crowding. When introducing the smallest nanochannels used in this work, initial trials with these systems exhibited lower experiment success rates. However, once the experimental protocol was tuned the “pass fraction” of experiments is comparable and high across all used nanochannel sizes. In other words smaller nanochannels do not intrinsically reduce the rate of successful measurements.”

2. How robust is the trained model in tolerance to the changes in measurement conditions? Such as changes in illumination, camera, channel size, or temperature. When re-training is necessary?

Our reply: We thank the reviewer for this important question. Our per-kymograph normalization and the probability-weighted property head in h-ViT provide tolerance to moderate changes in irradiated light intensity or changes in camera settings. However, changes in, for example, nanochannel cross-section or temperature alter the $D \rightarrow R_s$ and $iOC \rightarrow MW$ mappings. In those cases, it is necessary to either retrain on synthetic data matched to the new nanochannel geometry and noise, or to reuse the probability head and re-calibrate the property head. We have added a note in the Methods section stating that geometry changes generally warrant re-calibration/retraining of the h-ViT model that reads as follows:

“In practice, per-kymograph normalization and the probability-weighted property head make the model tolerant to moderate illumination or camera changes, so no retraining is required in those cases. By contrast, material changes in channel cross-section or temperature alter the $D \rightarrow R_s$ and $iOC \rightarrow MW$ mappings. Our default rule is therefore to (i) regenerate matched synthetic data and retrain end-

to-end, or (ii) freeze the probability head and re-calibrate the property head with a short geometry-specific calibration. We adopt option (ii) when the imaging statistics are stable and only the physical mapping shifts, and option (i) when noise statistics and geometry both change.”

3. Given the nanoscale channel size that requires highly purified samples for reliable measurement, and the lack of specificity other than molecular size, what could be the potential practical applications of the technology?

Our reply: Our more general response to this comment is identical to the one we wrote above in response to the Reviewers general question and the practical usefulness of the NSM method, which we don't reiterate here. Instead, here we focus on the aspect of specificity raised by the reviewer, where we agree that in "standard" NSM reported to date specificity comes from physical size and weight determination of single molecules diffusing in a nanochannel. However, as a first important point, we note that nanofluidic channels can be functionalized with analytes to engineer nanochannel surfaces to either prevent or enable specific binding (ACS Applied Materials Interfaces 2023, 15 (7), 10228-10239) like any open surface surface. Since we already in our seminal work have demonstrated that NSM also can resolve single molecules transiently bound to a surface (extended data figure 5 in Nature Methods 2022 Vol. 19 Issue 6 Pages 751-758), this means that affinity-binding-type single molecule experiments - possibly in sequence with non-affinity analysis when the molecules diffuse freely - are entirely possible and thus could provide an interesting (but yet unexplored) route to introduce chemical specificity. As a second (also still widely unexplored) potential of NSM we note the possibility to introduce orthogonal selectivity by exploiting the potential of nanofluidic concepts, such as geometric size exclusion/filtering or nanofluidic fractionation (Analytical Chemistry 2025, 97 (16), 8641-8653; Mao, P.; Fu, J., Nanofluidic Devices for Rapid Continuous-Flow Bioseparation. In Nanoproteomics: Methods and Protocols, Toms, S. A.; Weil, R. J., Eds. Humana Press: Totowa, NJ, 2011; pp 127-140.) in combination with NSM readout on the same chip.

To briefly address this important issue in the revised manuscript, we have added the following text at the end of the conclusions section:

"Looking forward, we envision the use of NSM in general for label-free determination of homogeneity, oligomeric state, and stoichiometry of biological molecular samples or for investigations of molecular interactions in solution. The specific concepts developed in this work open the door to such studies even in biologi-

cally highly relevant families of molecules in the sub-10 kDa size regime, such as cytokines, chemokines and peptide hormones, which have remained inaccessible to label-free single molecule microcopies due to their small sizes. Furthermore, as already demonstrated (Nature Methods 2022 Vol. 19 Issue 6 Pages 751-758), NSM can also be applied to characterize biological nanoparticles in terms of size and optical contrast, which in turn is linked to their refractive index and, as we predict, to their molecular cargo loading, such as for example mRNA. As a further forward-looking aspect, we note that specificity beyond molecular size and weight may be obtained either by introducing surface modifications inside nanofluidic channels by adapting surface chemistry developed for open surfaces (ACS Applied Materials Interfaces 2023, 15 (7), 10228-10239) that enables tailored specific binding of single molecules to engineered receptors or to explore the exciting possibility to introduce orthogonal selectivity by geometric size exclusion/filtering or nanofluidic fractionation (Analytical Chemistry 2025, 97 (16), 8641-8653; Mao, P.; Fu, J., Nanofluidic Devices for Rapid Continuous-Flow Bioseparation. In Nanoproteomics: Methods and Protocols, Toms, S. A.; Weil, R. J., Eds. Humana Press: Totowa, NJ, 2011; pp 127-140.) in combination with NSM readout on the same chip”

4. At the beginning of the introduction, some existing technologies are missing from the list of existing technologies, such as plasmonic scattering microscopy (Nature Methods 2020, V17, 1010-1017), evanescent scattering microscopy (Nature Communications, 2022, V13, 2298), and single protein oscillators (Nature Communications, 2020, V11, 4768).

Our reply: We thank the reviewer for the suggested additional references and we have added the suggested citations in the first paragraph of the Introduction and position NSM among label-free single-molecule methods that do not require surface immobilization during measurement as follows: *“...plasmonic scattering microscopy (PSM) (Nature Methods 2020, V17, 1010-1017), evanescent scattering microscopy (ESM) (Nature Communications, 2022, V13, 2298), single-protein oscillators (Nature Communications, 2020, V11, 4768)...*

...Within this landscape, NSM is positioned among label-free single-molecule approaches that do not require surface immobilization during measurement, enabling the characterization of freely diffusing analytes in solution while preserving native behavior.”

5. How to cross-validate the smaller molecule results that the traces of the molecule cannot be visually detected in the kymographs? Can simultaneous fluorescence

imaging be performed?

Our reply: We agree that simultaneous NSM-fluorescence imaging would be a great way of cross-validating the presence of molecules also in the sub 10 kDa regime. However, while this combination of methods in principle is possible to implement on our instrument, it is impossible to do (currently) for the small molecules at hand since they require photon fluxes that are at the limit of what our objective can handle without damage. Hence, the losses induced by simultaneous fluorescence imaging would push ourselves above that threshold if enough photons still are to be provided for NSM of these small molecules in the smallest nanochannels. Therefore, for "cross-validation" in this study we rely on the stringent negative control experiments with only PBS buffer in the system and the convergence to the CRLB with increasing N as orthogonal validation that the signal indeed originates from molecules rather than structured noise.

6. Page 12, last paragraph, line 7, "5 µg/ml mM" should be "5 µg/ml".

Our reply: We have corrected "5 µg/ml mM" to "5 µg/ml".

Reviewer #3:

Reviewer #3 (Remarks on code availability):

The code seems good and working fine!

Our reply: We appreciate the positive feedback that the code works as intended.

References

1. B. Špačková, *et al.*, *Nature Methods* **19**, 751 (2022).
2. M. Ghahremani, *et al.*, *2024 IEEE/CVF Conference on Computer Vision and Pattern Recognition (CVPR)* (2024), pp. 11513–11523.
3. C. R. Harris, *et al.*, *Nature* **585**, 357 (2020).
4. P. Virtanen, *et al.*, *Nature Methods* **17**, 261 (2020).
5. J. Lin, *IEEE Transactions on Information Theory* **37**, 145 (1991).
6. S. Kullback, R. A. Leibler, *Annals of Mathematical Statistics* **22**, 79 (1951).
7. C. Villani, *Optimal Transport: Old and New*, vol. 338 of *Grundlehren der mathematischen Wissenschaften* (Springer, 2009).
8. A. Kolmogorov, *Giornale dell'Istituto Italiano degli Attuari* **4**, 83 (1933).
9. N. Smirnov, *Annals of Mathematical Statistics* **19**, 279 (1948).
10. M. Heusel, H. Ramsauer, T. Unterthiner, B. Nessler, S. Hochreiter, *Advances in Neural Information Processing Systems* (2017), vol. 30.
11. A. Gretton, K. Borgwardt, M. Rasch, B. Schölkopf, A. Smola, *Journal of Machine Learning Research* **13**, 723 (2012).